# Mechanistic insights into the non-equilibrium thermodynamics of nitrogen fixation via acoustic cavitation

Xuelei Pan [1,3], Davide Bernardo Preso [1,3], Qian Liu[2,3], Lucia Mullings [1], Mohamad Salimi[1], Yi Qin[1], Pankaj S. Sinhmar[1] & James Kwan [1] ✉

Non-equilibrium reaction environments offer a route to bypass the thermodynamic constraints that limit conventional nitrogen fixation, yet such conditions remain inaccessible in traditional thermal systems. Here, we show that rapid activation-quenching chemistry inside cavitation bubbles provides a viable non-equilibrium pathway for nitrogen fixation. The violent collapse of ultrasound-driven bubbles generates an intense temperature pulse that enables direct nitrogen activation and subsequent redox chemistry within a transient gas phase microreactor. Nitrogen-containing products are produced with tuneable rates and selectivity controlled by feed gas composition, cavitation dynamics, and solution properties. Introduced cavitation nuclei lower the cavitation threshold and improve collapse reproducibility, while noble-gas doping modulates collapse temperatures and shifts nitrate-nitrite distributions through enhancing the involvement of water-derived species. Isotopic labelling and single-bubble modelling indicate that nitrogen reaction proceeds predominantly through gas-phase pathways during collapse, which can be described by a dynamic thermodynamic model within a temperature pulse. These findings establish cavitation-driven non-equilibrium thermal cycling as a distinct mechanism for nitrogen fixation and underscore the broader potential of transient thermal microenvironments for chemical synthesis.

Nitrogen fixation, the conversion of inert atmospheric nitrogen (N$_2$) into reactive nitrogen compounds, underpins global agriculture and chemical manufacturing[1–3]. Artificial nitrogen fixation is predominantly achieved through the Haber–Bosch process[4,5], which produces ammonia from nitrogen and hydrogen under high temperatures and pressures. In the industrial nitrogen cycle, ammonia is further oxidised to produce nitrate[6], a key component of fertilisers and other nitrogen-based chemicals[7]. Despite nitrate being a major form of industrial nitrogen products[8], the direct oxidation of nitrogen molecules remains a fundamental challenge in chemistry and catalysis[7,9,10]. The primary difficulty lies in overcoming the high dissociation energy of the nitrogen triple bond[11] (≈9.8 eV at 298 K). Direct oxidation typically requires temperatures exceeding 1500 K, yet the reaction rate remains low[12], and further increases in temperature are not practical due to material instability under such conditions. In contrast, nitrogen reduction is thermodynamically exothermic, meaning high temperatures do not favour the forward reaction due to the constraints imposed by chemical equilibrium, which leads to the high-pressure system[13]. Therefore, the central challenge in nitrogen fixation lies in reconciling the conflicting requirements of achieving sufficient activation energy while maintaining reaction conditions that favour nitrogen product formation. This delicate balance between chemical

[1]Department of Engineering Science, University of Oxford, Parks Road, Oxford, United Kingdom. [2]Hoffmann Institute of Advanced Materials, Shenzhen Polytechnic University, 7098 Liuxian Boulevard, Shenzhen, China. [3]These authors contributed equally: Xuelei Pan, Davide Bernardo Preso, and Qian Liu. ✉e-mail: james.kwan@balliol.ox.ac.uk

kinetics and thermodynamic limitations remains a major bottleneck in the development of efficient and sustainable nitrogen fixation technologies.

Recently, programmable heating and quenching strategies have been developed to regulate reaction pathways, aiming to facilitate nitrogen activation while preserving catalyst stability[14]. These electrothermal systems offer controlled temperature modulation, with ramping and cooling rates reaching up to $10^4$ K s$^{-1}$. While this represents a significant advance over conventional heating methods, such rates are only sufficient to influence reactions on the millisecond timescale. Consequently, they are unable to selectively control sequences of elementary reaction steps that occur on much shorter timescales. Achieving selective control, therefore, requires both fast temperature oscillation and access to high temperatures that are inaccessible to conventional reactors.

Acoustic cavitation provides such conditions. Under ultrasound irradiation, gas–vapour bubbles nucleate, expand, and undergo inertial collapse in liquids[15,16]. The collapse of these cavitation bubbles generates localised "hot spots" with transient high temperature and pressure pulse[17,18]. These non-equilibrium conditions enable molecular activation and product formation steps to occur in distinct temporal regimes, offering a fundamentally different route to nitrogen fixation. A schematic illustration of the nitrogen fixation process via cavitation is shown in Fig. 1a. The cavitation bubbles behave as confined microreactors. Upon violent inertial collapse, the temperature and pressure inside the bubbles rise dramatically, driven by the nearly adiabatic compression of the gas. These conditions trigger the thermolytic dissociation of gas-phase molecules and generate a range of reactive radical species (e.g., N, O, H, OH). Subsequent recombination reactions yield nitrogen-containing products, which are rapidly quenched and

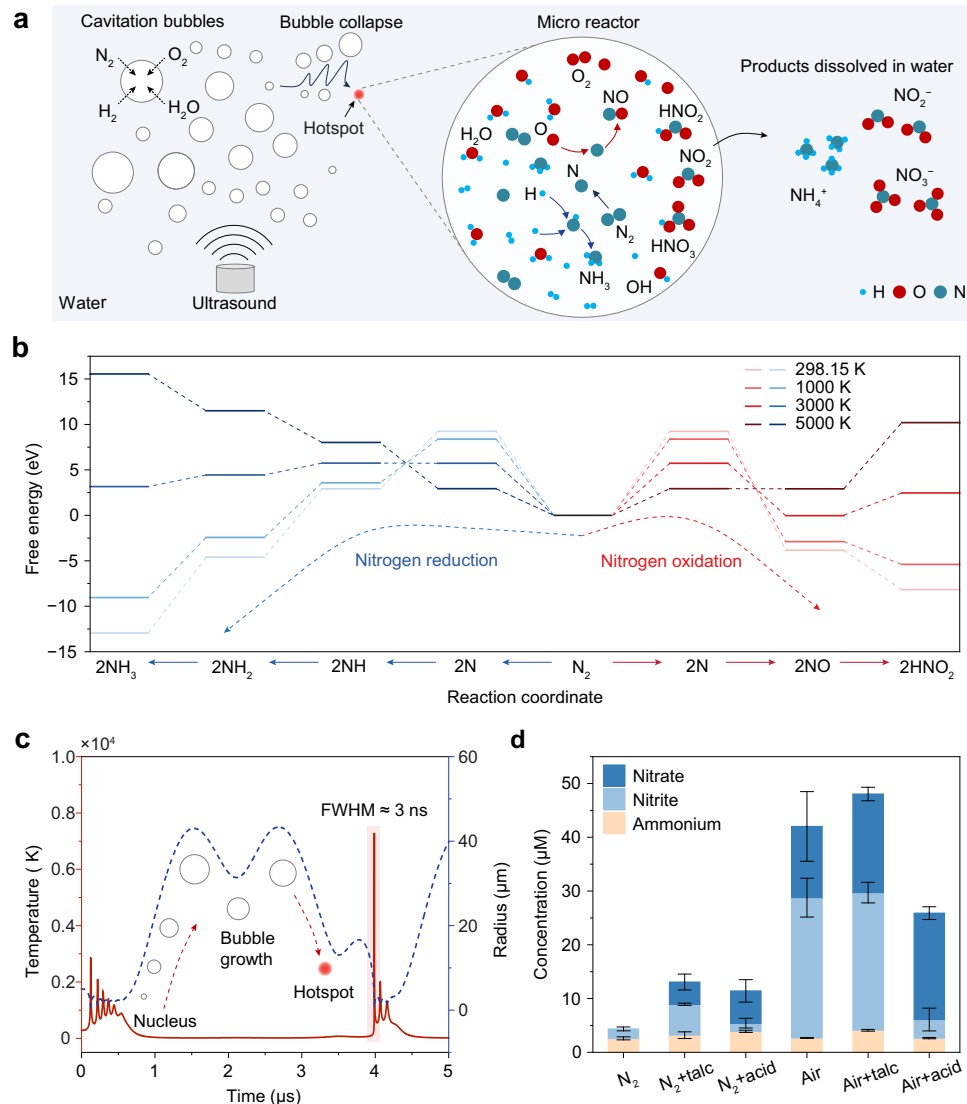

**Fig. 1 | Nitrogen fixation reaction in cavitation bubbles. a** Schematic illustration of nitrogen fixation via acoustic cavitation, where ultrasound induces the formation and collapse of gas microbubbles. The right part shows the conceptual diagram of radical generation and recombination reactions inside a collapsing bubble. The light blue background indicates water environments; grey circles represent gas bubbles. **b** The left part shows Gibbs free energy profiles for nitrogen reduction to ammonia via stepwise addition of H radicals at four representative temperatures. The right part shows the corresponding energy profiles for nitrogen oxidation,

beginning with NO formation, followed by reaction with OH radicals to produce HNO$_2$. **c** Simulated dynamics of a nitrogen–water vapour bubble under ultrasonic irradiation. The temperature spike exhibits a typical full width at half maximum (FWHM) of ≈3 ns. **d** The comparison of products from pure nitrogen and air in water, acidic solution (0.05 M H$_2$SO$_4$) and water with talc (1 mg mL$^{-1}$). Each data point represents the mean of three independent measurements conducted under identical conditions, see Source Data; error bars indicate the standard deviation. Source data are provided as a Source Data file.

transferred into the surrounding liquid, allowing accumulation in solution. This phenomenon represents the chemical effects of ultrasound, collectively referred to as sonochemistry.

Although nitrogen fixation via acoustic cavitation has been previously observed[19–23], the mechanistic basis and controllability remain elusive[24]. As a result, this phenomenon has often been regarded as a secondary effect of sonochemistry rather than a viable pathway for nitrogen fixation[25]. Early pioneering studies reported the formation of nitrite and nitrate in air-saturated water under ultrasound, with most attributing the process to a high-temperature combustion-like pathway[26], namely, the Zeldovich mechanism[27]. However, this explanation fails to account for the observed non-stoichiometric product distributions and the significant role of water in the sonochemical reaction system. Critically, no systematic strategies have been proposed to modulate the reaction pathway or control the product distribution. In contrast, the production of ammonia via cavitation has been scarcely reported in the literature[23], likely due to the assumption that ammonia synthesis is incompatible with high-temperature combustion conditions.

In this work, we establish a dynamic thermodynamic framework to elucidate the chemistry occurring during bubble collapse. We identify the physical origin of inertial cavitation as the key factor governing nitrogen activation and demonstrate that nitrogen oxidation and reduction pathways can coexist under the same sonochemical conditions. This behaviour contrasts with the disproportionation mechanisms recently proposed for microdroplets[28]. We attribute this unique reactivity to the combination of transient high temperatures, sufficient to activate molecular nitrogen, followed by rapid quenching that selectively stabilises reaction intermediates and products. This rapid thermal cycling plays a decisive role in directing the reaction pathway. Furthermore, the gas-phase reaction products formed within the collapsing bubbles diffuse into the surrounding liquid phase, effectively shifting the chemical equilibrium and enabling continued product accumulation.

## Results

### Direct gas phase reaction in collapsing cavitation bubbles

To model the chemistry in cavitation bubbles, we consider the direct radical reaction pathway, where molecular nitrogen undergoes direct dissociation. The associated free energy landscape reveals that nitrogen dissociation is strongly endothermic, while subsequent steps, such as the formation of nitrous acid or ammonia, are energetically downhill (Fig. 1b). At $5000\,K$, $N\equiv N$ bond dissociation becomes accessible, whereas nearly all steps, including product formation, become thermodynamically uphill. Stable nitrogen products, therefore, require a rapidly oscillating thermal profile: transient high temperature to activate $N_2$, followed by fast quenching to stabilise intermediates. This dynamic temperature cycling is intrinsic to inertial bubble collapse and cannot be achieved in conventional steady-state thermal systems. For example, the collapse of nitrogen–water vapour bubbles can produce peak temperatures approaching $5000\,K$ or higher (Fig. 1c). Importantly, this thermal pulse occurs within $\approx 3\,ns$, corresponding to a heating/cooling rate on the order of $10^{12}\,K\,s^{-1}$. Herein, different elementary reaction steps are initiated at distinct stages of this transient thermal cycle. By enabling these processes to occur sequentially under optimised thermal conditions, cavitation offers an opportunity to overcome both kinetic and thermodynamic constraints that limit conventional nitrogen fixation technologies.

To evaluate nitrogen fixation under these conditions, we investigated two closed reaction systems (Supplementary Fig. 1): water saturated with pure nitrogen or air. The measured UV-vis spectra of nitrogen products after 30 min are shown in Supplementary Fig. 3 (Calibration curves present in Supplementary Fig. 2). Pure $N_2$ produced only trace amounts of fixed nitrogen, whereas air led to substantially higher yields ($42\,\mu M$), indicating that oxygen is essential for

initiating efficient N–O chemistry (Fig. 1d). However, the air-saturated system showed variability due to stochastic cavitation behaviour.

To stabilise cavitation, talc (magnesium silicate, $10\,\mu m$ particle size, see structural characterisation in Supplementary Fig. 4) was introduced as an inert cavitation agent. The cavitation activity with talc is due to the transition from stochastic nucleation to controlled nucleation of bubbles. As shown in Supplementary Fig. 5, adding talc to water lowers the cavitation threshold and enables controllable bubble collapse. As a result, talc increased the total nitrogen fixation for both pure nitrogen and air compared to the pure water system. For reaction with gas, the incorporation of talc specifically increased nitrate production without decreasing nitrite production. We also found that by adjusting the pH with sulphuric acid, nitrate became the dominant product for both cases, while total nitrogen fixation decreased for the air reaction. This indicates that liquid properties also play a role in the nitrogen fixation reaction.

These comparative results suggest that the primary reaction mechanism is driven by gas-phase interactions inside the cavitation bubbles, rather than reactions between dissolved gases and water in the bulk liquid. This is mainly concluded from the low nitrogen fixation of pure nitrogen reaction, which lacks reactive oxygen-containing species required to initiate efficient radical chemistry. Although ref. [29] has proposed that hydroxyl (OH) radicals generated from water contribute significantly to liquid-phase nitrogen fixation, our findings indicate that gas-phase reactions during bubble collapse are the dominant pathway in this system. This interpretation is further supported by the enhanced nitrogen fixation observed upon the addition of talc particles. Talc acts as a cavitation nucleus source, increasing the number of stable bubble nucleation sites and thereby promoting more frequent and intense bubble collapse events. This leads to increased hotspot formation and radical generation, resulting in higher nitrogen fixation yield.

Conversely, introducing sulphuric acid leads to a decrease in total nitrogen fixation when air is used as the feed gas. This behaviour is attributed to the salting-out effect, in which the presence of ionic species reduces the solubility of both $N_2$ and $O_2$ in water, thereby lowering the gas content within bubbles and reducing reaction yield[30]. Because the air-based system relies more strongly on gas incorporation and radical oxidation pathways, its reaction rate is more sensitive to this reduction in gas solubility. The relative increase of nitrate concentration suggests that nitrate formation is influenced by solution-phase oxidation processes occurring after bubble collapse[19]. Taken together, these results reinforce the conclusion that the microenvironment inside collapsing cavitation bubbles determines both the extent and the selectivity of nitrogen fixation, while subsequent solution chemistry modulates the relative distribution of oxidised nitrogen products.

### Reaction regulated by cavitation and gas–liquid conditions

Following the above analysis, it is evident that the amount of ammonium produced remains consistently low, even under pure nitrogen conditions. This contradicts conventional nitrogen fixation pathways due to the more moderate reaction conditions required for nitrogen reduction compared to nitrogen oxidation. However, this observation suggests that the direct reaction between nitrogen and water vapour is not a favourable pathway for ammonium synthesis. Motivated by these findings, we further explored nitrogen fixation using different gas mixtures, focusing on two closed reaction systems: water saturated with either nitrogen–oxygen or nitrogen–hydrogen gas mixtures. We then studied product distributions as a function of nitrogen concentration, acoustic pressure of ultrasound, and reaction time (UV-vis spectra are shown in Supplementary Figs. 6, 7).

For nitrogen–oxygen mixtures, the optimal nitrogen fixation was observed at a nitrogen concentration of $\approx 79\%$ (air) (Fig. 2a). Interestingly, a 50% nitrogen concentration did not show the highest yield,

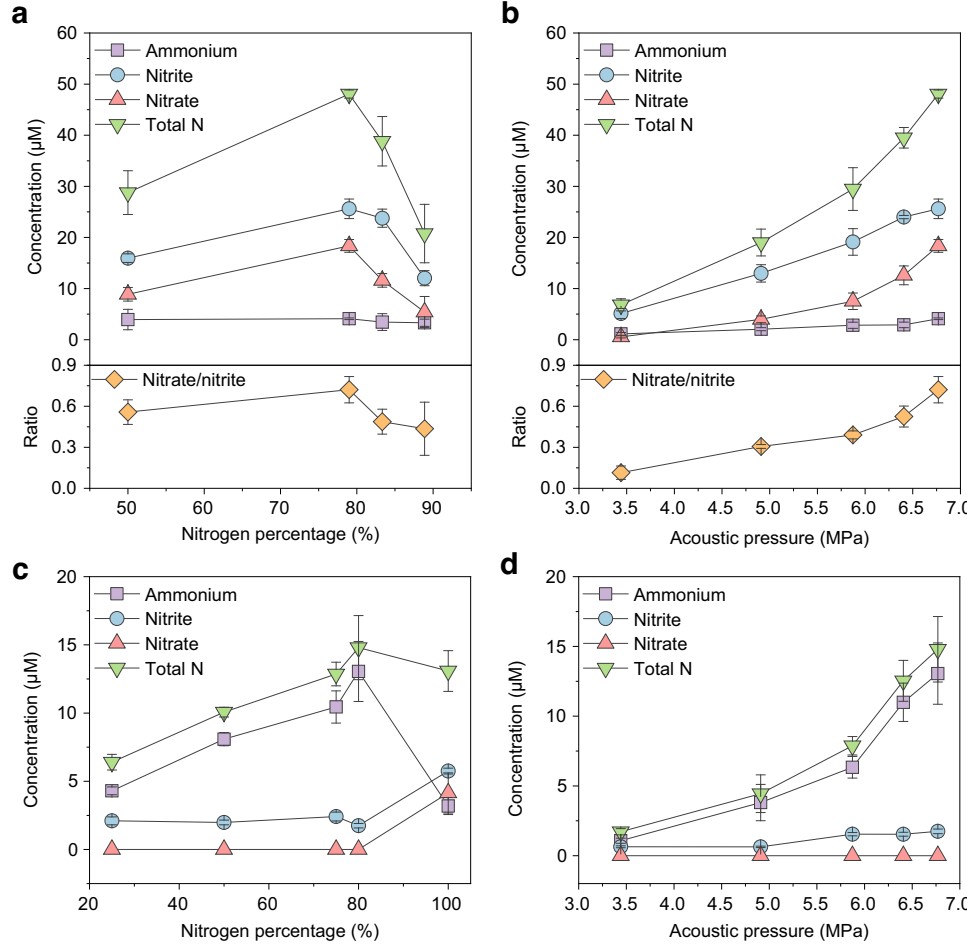

**Fig. 2 | Nitrogen fixation products as a function of feed gas composition and acoustic pressure. a** Product concentrations as a function of nitrogen percentage in $N_2$–$O_2$ gas mixtures. The bottom plot shows the ratio of nitrate-to-nitrite. All experiments were conducted with 1 mg mL$^{-1}$ talc, duty cycle is 6.1% with 100 cycles per 2 ms burst, in water for 30 min unless otherwise specified. **b** Product concentrations as a function of acoustic pressure after 30 min of reaction with air as the feed gas. The bottom plot shows the ratio of nitrate-to-nitrite. **c** Product concentrations as a function of nitrogen percentage in $N_2$–$H_2$ gas mixtures. **d** Product concentrations as a function of acoustic pressure after 30 min of reaction with 80% $N_2$–20% $H_2$ as the feed gas. **a**–**d** Each data point represents the mean of three independent measurements, see Source Data; error bars indicate the standard deviation. Source data are provided as a Source Data file.

despite being stoichiometric for NO formation if a direct 1:1 reaction between nitrogen and oxygen were assumed. Previous studies, including computational analyses based on the Zeldovich mechanism, have shown that the maximum $NO_x$ production occurs at around 70% nitrogen, while experimental results are different[22]. According to the Zeldovich mechanism, the initial step involves oxygen molecule dissociation to form O radicals, which then readily react with $N_2$ to produce NO and N radicals. This explanation aligns with our observations, as the equilibrium formation of reactive O species is favoured at specific nitrogen-to-oxygen ratios. Nevertheless, few studies have conducted in-depth investigations into this deviation under non-equilibrium conditions such as those created by acoustic cavitation. This intriguing phenomenon will be discussed in greater detail in subsequent sections.

After identifying the optimal nitrogen-to-oxygen ratio, we examined product distributions under varying acoustic pressures. When air was used as the feed gas (Fig. 2b), the nitrite concentration initially increased almost linearly with increasing acoustic pressure and then reached a plateau. By contrast, the nitrate concentration continued to rise more sharply across the entire pressure range. This divergence suggests that higher acoustic pressures not only enhance primary nitrogen activation but also increasingly favour nitrate formation. Such nitrate enhancement may arise from direct oxidation pathways

inside the collapsing bubble or post-collapse liquid-phase oxidation of nitrite. To further test this hypothesis, we monitored time-dependent nitrogen fixation in a closed, air-saturated system (Supplementary Fig. 8). The total amount of fixed nitrogen exhibited a linear increase with reaction time, indicating effective reaction kinetics in a closed system, not limited by dissolved gas concentration. Notably, while the accumulation rate of nitrite gradually slowed over time, the nitrate concentration continued to rise steadily. This temporal divergence supports a sequential mechanism in which nitrite serves as a precursor to nitrate, with nitrate formation partially governed by secondary oxidation processes occurring after bubble collapse.

To better visualise this trend, the nitrate-to-nitrite ratio was calculated and plotted. A nitrogen–oxygen mixture containing 79% nitrogen yielded the highest nitrate-to-nitrite ratio of 0.72, which further increased to 0.99 with prolonged reaction time (Supplementary Fig. 8h). More significantly, the nitrate-to-nitrite ratio exhibited a clear positive correlation with acoustic pressure, rising sharply from 0.11 to 0.72. This behaviour may be attributed to two main factors: first, stronger cavitation activity directly enhances nitrate formation during bubble collapse; second, increased reaction time in the closed system leads to accumulation of OH radicals, which facilitates the secondary oxidation of nitrite to nitrate in water. OH radicals are from the pyrolysis of water and can form $H_2O_2$ in water. Supplementary Fig. 9

presents the $H_2O_2$ concentrations under different feed gases. As a benchmark, pure argon yielded the highest concentration of $H_2O_2$, while significantly lower concentrations were observed with nitrogen and nitrogen-containing mixtures. Although differences in thermal conductivity among the gas mixtures may influence hotspot temperatures, the substantial reduction in $H_2O_2$ suggests that OH radicals are being consumed by reactions with nitrogen species.

In the nitrogen–hydrogen mixture system (Fig. 2c, d), the main product was ammonium, while negligible nitrate was detected. The maximum nitrogen fixation was achieved at 80% nitrogen concentration (Fig. 2c). Compared to the nitrogen–oxygen system, where ammonium production was minimal, the clear production of ammonium in the nitrogen–hydrogen mixture suggests direct N–H coupling in the gas phase. Interestingly, the stoichiometric ratio for ammonia synthesis (25% nitrogen and 75% hydrogen) exhibited much lower ammonium production, indicating that the reaction does not simply follow a stoichiometric mechanism but is governed by the complex processes inside the collapsing bubbles. For time-dependent experiments with 80% nitrogen, ammonium concentration increased linearly with reaction time (Supplementary Fig. 8i). A similar trend was observed in acoustic pressure-dependent experiments (Fig. 2d), further confirming the high selectivity of ammonium production under these conditions. It is noteworthy that the total nitrogen fixation in the nitrogen–hydrogen system was less than half that observed for the nitrogen–oxygen system, which reflects differences in the reaction kinetics between nitrite and ammonium formation.

To further evaluate system performance, we compared the production normalised to the energy input (Supplementary Fig. 10). In the nitrogen–oxygen system, nitrogen fixation continuously increases with the pulse energy (100 cycles), reaching a maximum yield of $0.18 \, \mu mol \, J^{-1}$. Nitrite formation plateaued at a pulse energy of 11.09 µJ, corresponding to a yield of $0.098 \, \mu mol \, J^{-1}$, whereas nitrate yields continued to increase with higher pulse energy. In the nitrogen–hydrogen system, ammonium formation exhibited a maximum at 12.38 µJ per pulse, giving a peak yield of $0.048 \, \mu mol \, J^{-1}$, while nitrite concentrations decreased progressively with increasing pulse energy.

In addition to varying the acoustic pressure, we also investigated the role of cavitation agents in sonochemical nitrogen fixation (Supplementary Fig. 11). As previously demonstrated, the addition of talc enhances cavitation activity. To systematically assess its impact, we monitored product distributions across a range of talc concentrations ($0-4 \, mg \, mL^{-1}$). In the nitrogen–oxygen system, reactions conducted without talc showed large fluctuations in total nitrogen fixation, comparable to those observed at 1 and $2 \, mg \, mL^{-1}$ talc. Interestingly, nitrate production decreased with increasing talc concentration, while nitrite levels remained relatively constant. This suggests that cavitation activity affects not only the overall extent of nitrogen fixation but also the selectivity between nitrite and nitrate formation. In the nitrogen–hydrogen system, maximum ammonium production was achieved at a talc concentration of $1 \, mg \, mL^{-1}$. In the absence of talc, nitrogen fixation was significantly lower, and increasing the talc concentration beyond $1 \, mg \, mL^{-1}$ did not lead to further improvement in ammonium yield. These findings indicate that both nitrate and nitrite production can be modulated by tuning bubble dynamics via cavitation agents. The addition of talc promotes inertial collapse by reducing the inertial cavitation pressure threshold. This alters bubble-collapse dynamics, thereby suppressing direct nitrate formation, which requires longer sequential gas-phase reaction pathways. These results support the notion that nitrate can be formed directly within the bubbles during collapse, in addition to subsequent reactions in the liquid phase.

Overall, these findings highlight the complex interplay between primary gas-phase reactions occurring within collapsing cavitation bubbles and secondary liquid-phase reactions driven by reactive oxygen species. They underscore the critical role of cavitation activity, gas-phase composition, and transient thermal dynamics in modulating nitrogen fixation pathways under sonochemical conditions.

In the closed reaction system, our primary focus was on analysing liquid-phase products; however, gas-phase species are also critically important, particularly under closed conditions where accumulation and secondary reactions may occur. It is well-established that ultrasound treatment leads to the production of hydrogen through the recombination of H generated by water splitting. Using argon as the feed gas, hydrogen production was confirmed via mass spectrometry (Supplementary Fig. 12). Similar hydrogen formation was also observed in nitrogen-saturated systems (Fig. 3a), validating that water splitting occurs regardless of the feed gas composition. When a mixed $H_2O/D_2O$ solution was employed, the detection of HD and $D_2$ signals further confirmed active radical recombination and water splitting.

These results indicate that sonication always induces water splitting, contributing $H_2$ and $O_2$ to the system. These gaseous products may influence the local reaction environment inside cavitation bubbles, potentially altering product selectivity and formation rates. To further examine this, we performed reactions under continuous gas flow conditions (Fig. 3b, c, Supplementary Fig. 13). The continuous introduction of gas nuclei into the liquid helps maintain stable and efficient cavitation. With pure nitrogen as the feed gas, nearly equal amounts of nitrite and nitrate were produced, accompanied by a gradual increase in ammonium concentration over time (Supplementary Fig. 14a). In this setup, nitrogen can only react with water-derived radicals, indicating that OH and H participate in nitrogen fixation, although their reaction rate ($0.27 \, \mu M \, min^{-1}$) is substantially lower than that observed for reactions involving oxygen or hydrogen gas. Interestingly, the ratio of nitrate to nitrite is around 1.0 (Fig. 3b), which shows different behaviour to direct reaction with oxygen.

When air was used as the feed gas (Fig. 3b), the total nitrogen fixation rate increased significantly to $1.69 \, \mu M \, min^{-1}$. Both nitrite and nitrate concentrations showed strong linearity over time, while ammonium remained negligible (Supplementary Fig. 14b). We find that the ratio of nitrate to nitrite slightly increases from 0.32 to 0.53. The continuous refreshment of the gas stream helped stabilise the system and maintain the nitrate to nitrite ratio, which otherwise might be suppressed by secondary reactions in a closed system. In contrast, nitrogen fixation rate in the closed system is $1.38 \, \mu M \, min^{-1}$ and nitrate to nitrite ratio is around 0.7 (Supplementary Fig. 8g, h), which demonstrates that the flow system delivers stable reaction kinetics. We further explored the possibility of the involvement of the products in further reaction. We flowed hydrogen gas into 10 mM potassium nitrate, and the reduction products are negligible ($\approx 1 \, \mu M$ after 30 min reaction), which can be attributed to the sluggish radical reaction in liquid (Supplementary Fig. 15).

Herein, we further investigated strategies to selectively promote the formation of nitrogen-containing products. In Fig. 3c, Supplementary Fig. 16, we show that in the continuous-flow system, nitrogen–oxygen mixtures selectively produced nitrate in acidic solution at an average rate of $1.03 \, \mu M \, min^{-1}$, with nitrite remaining negligible. In contrast, selective ammonia formation requires the introduction of hydrogen gas. Nitrogen–hydrogen mixtures produced ammonium as the dominant product, with a rate of $0.45 \, \mu M \, min^{-1}$. This ammonium production rate is comparable to that obtained in the closed system ($0.41 \, \mu M \, min^{-1}$) (Supplementary Fig. 8i) but exhibits substantially improved linearity with reaction time under continuous-flow operation, indicating enhanced stability and reproducibility. We also measured the ammonium concentration in the acid trap at downstream remained below the detection limit (<0.05 ppm) even after 2 h of continuous operation indicating that no measurable ammonia is released into the gas phase during the reaction (Supplementary Fig. 17). In both cases, product concentrations were linearly

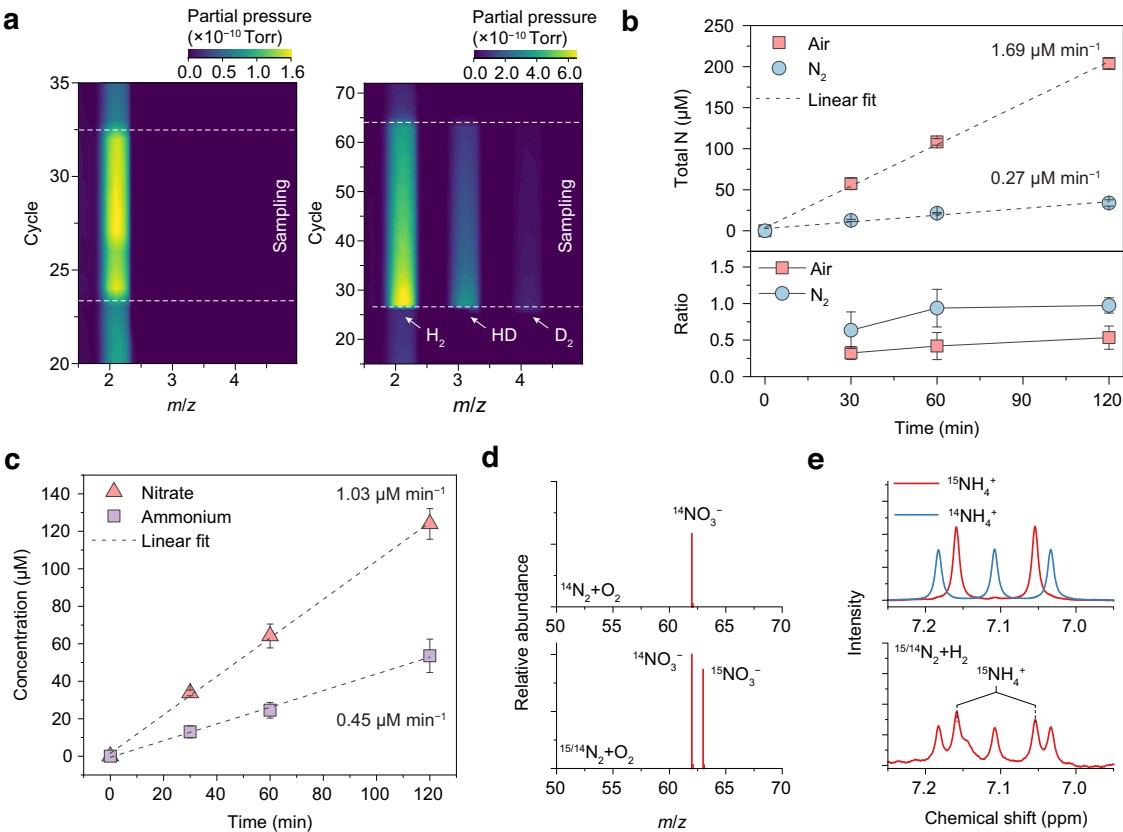

**Fig. 3 | Nitrogen fixation in the continuous-flow sonochemical system.**
**a** Contour plots of mass spectra for products with $N_2$ as the feed gas in $H_2O$ (left) and in mixed $D_2O/H_2O$ solutions (volume ratio 3:2) (right). The sampling region is between two dashed lines. **b** Total fixed nitrogen and nitrate-to-nitrite ratios from reactions using pure $N_2$ and air as feed gases in water with 1 mg mL$^{-1}$ talc. The coefficient of determination $R^2$ for the linear fitting of air and $N_2$ is 0.998 and 0.970, respectively. Ultrasound was applied with a 6.1% duty cycle (100 cycles per 2 ms burst). **c** Nitrate production from air in acidic solution (0.05 M $H_2SO_4$) and

ammonium production from an $N_2$–$H_2$ mixture (80% $N_2$) in neutral water, both with 1 mg mL$^{-1}$ talc as the cavitation agent. The $R^2$ for linear fitting of nitrate and ammonium production is 0.999 and 0.998, respectively. **d** Mass spectra of nitrate produced using $N_2$–$O_2$ and $^{15/14}N_2$–$O_2$ gas mixtures. **e** $^1H$ NMR spectra of $^{14}NH_4^+$ and $^{15}NH_4^+$ reference standards and of the products formed using a $^{15/14}N_2$–$H_2$ mixture. **b**, **c** Each data point represents the mean of three independent measurements, see Source Data; error bars indicate the standard deviation. Source data are provided as a Source Data file.

dependent on time, further confirming the validity and selectivity of the sonochemical nitrogen fixation reactions. These findings demonstrate that by altering the feed gas composition, our system can selectively synthesise either nitrate or ammonium. We also checked the morphology of talc after 2 h reaction (Supplementary Fig. 4c) and observed no changes to the original structure. The element mapping of talc before and after a 2 h reaction shows similar element distribution and Si/Mg ratio (Supplementary Fig. 18), demonstrating its stability as a cavitation agent.

We then employed $^{15}N$ isotopic labelling to confirm the involvement of molecular nitrogen in the reaction. In the oxidation pathway using a $^{15}N_2$ gas mixture, mass spectrometry clearly revealed a signal at $m/z = 63$, corresponding to $^{15}NO_3^-$ (Fig. 3d). For the reduction pathway, $^1H$ NMR spectra showed characteristic doublet peaks at 7.05 and 7.16 ppm, consistent with $^{15}NH_4^+$ (Fig. 3e). These results verify that the observed nitrogen-containing products originate directly from molecular nitrogen.

From the above experiments, we draw three principal conclusions: First, gas-phase reactions dominate sonochemical nitrogen fixation, occurring within collapsing cavitation bubbles where high temperatures favour molecular activation. Second, radicals generated from water, such as OH and H, can participate in nitrogen fixation, but their contribution is significantly less effective than that of molecular oxygen interactions. However, it influences the pathway and results in product distributions that deviate from expected stoichiometric ratios. Third, the primary reaction only happens in the gas phase,

making the products separate in liquid, which inhibits the products' decomposition.

## Reaction kinetics analysis
Having established the general trends in nitrogen fixation rates, we next investigated the reaction kinetics associated with bubble collapse. We first investigated reactions with both nitrogen–oxygen and nitrogen–hydrogen mixtures using different ultrasound burst periods (Supplementary Fig. 19). Optimal nitrogen fixation was observed at a 122 μs burst period (100 cycles), with the nitrate-to-nitrite ratio remaining constant at ≈0.4. The optimal fixation likely corresponds to the optimal operating state of the transducer, as previously reported in hydrogen production studies[31]. The invariance of the nitrate-to-nitrite ratio across burst periods strongly suggests that product formation occurs within a single bubble collapse event, on the scale of just a few acoustic cycles, and that changes in the burst period do not significantly affect the cavitation dynamics. Therefore, in evaluating the reaction mechanism, we should focus on the conditions during a single temperature spike.

To further probe the role of collapse temperature, we introduced noble gas argon (Ar), which is known to enhance hotspot temperatures during cavitation by reducing heat capacity within bubbles. Increasing the Ar fraction in air mixtures led to a pronounced rise in total nitrogen fixation, reaching a maximum at 50% Ar (Fig. 4a, Supplementary Fig. 20a). Simultaneously, the nitrate-to-nitrite ratio steadily increased from 0.32 to 0.62 with increasing Ar content. In contrast, ammonium

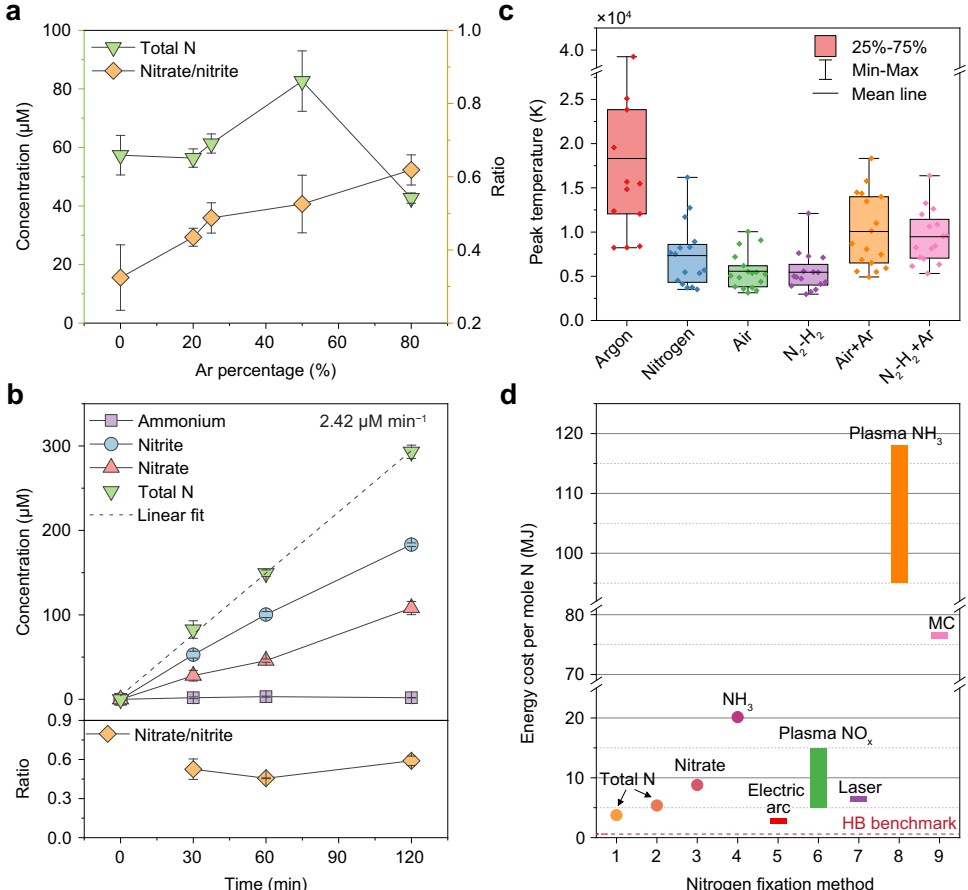

**Fig. 4 | Sonochemical nitrogen fixation rate and energy cost. a** Total nitrogen fixation and the corresponding nitrate-to-nitrite ratio as a function of Ar content in air–Ar gas mixtures. **b** Total fixed nitrogen and nitrate-to-nitrite ratios from reactions using 50% air and 50% Ar as feed gases in water with 1 mg mL$^{-1}$ talc. The $R^2$ value for linear fitting is 0.998. **a**, **b** Each data point represents the mean of three independent measurements, see Source Data; error bars indicate the standard deviation. **c** Simulated peak temperatures during single-bubble collapse events for various gas compositions, illustrating the temperature effect of gas components. Boxes indicate the interquartile range, the central line denotes the mean value, and whiskers represent the full minimum–maximum span; for individual data points, see Source Data. **d** The plot of the energy cost of different methods. The dashed line indicates the benchmark energy cost of the Haber–Bosch (HB) process (0.5–0.6 MJ mol$^{-1}$). Data points 1–4 represent results from this work (see Table 1): Point 1 is 50% air + 50% Ar in water, point 2 is air in water, point 3 is air in 0.05 M H$_2$SO$_4$, point 4 is 80% N$_2$ + 20% H$_2$ in water. The bars represent the typical range of energy cost from the literature. Method 5 corresponds to the electric arc (Birkeland–Eyde) method[32]. Method 6 represents plasma-based nitrogen oxidation[33]. Method 7 refers to the laser bubbling in liquid method[24]. Method 8 shows plasma-based nitrogen reduction[41]. Method 9 represents the mechanochemical (MC) nitrogen reduction method[42]. Source data are provided as a Source Data file.

concentrations in N$_2$–H$_2$ mixtures remained nearly constant upon Ar addition (Supplementary Fig. 20b). Meanwhile, both nitrate and nitrite concentrations continued to rise. However, when the Ar proportion reached 80%, nitrite became the dominant product, accompanied by a noticeable decrease in ammonia production.

This confirms that the nitrate-to-nitrite ratio is a sensitive indicator of the reaction mechanism tuned by cavitation conditions, particularly the collapse temperature. The shift in product distribution in the Ar-containing nitrogen–hydrogen system suggests a competitive interaction between hydrogen gas and water vapour. At elevated collapse temperatures, enhanced thermolysis of water vapour generates reactive OH/O radicals, which can outcompete molecular hydrogen in reacting with nitrogen species. Consequently, oxidised nitrogen products are favoured even in the absence of an external oxygen source. A similar effect can thus be inferred in the Ar-containing nitrogen–oxygen system, where the monotonically increasing nitrate-to-nitrite ratio with rising Ar content suggests increased participation of water-derived species in the reaction. This shift in mechanism likely leads to reduced overall nitrogen fixation yield, as more reactive nitrogen is diverted toward different pathways. The

results thus indicate that Ar not only raises the collapse temperature but also alters the relative contribution of competing reactants, modulating both product selectivity and total yield.

In the long-term reaction (Fig. 4b, Supplementary Fig. 21), the air–argon mixture maintained strong reaction kinetics, achieving a total nitrogen fixation rate of 2.42 µM min$^{-1}$, corresponding to a yield rate of 5.95 µmol h$^{-1}$ in a 2.5 mL vial. Both nitrate and nitrite concentrations increased linearly, with individual fixation rates of 0.89 and 1.51 µM min$^{-1}$, respectively. The nitrate-to-nitrite ratio stabilised around 0.6, slightly higher than that observed in the pure air reaction, indicating enhanced kinetics for nitrate production. However, a distinct trend was observed in the nitrogen–argon reaction system. As shown in Supplementary Fig. 22, the total nitrogen fixation rate reached 0.77 µM min$^{-1}$, nearly three times that of pure N$_2$. Meanwhile, the nitrate-to-nitrite ratio increased from 0.3 to 0.6 over 120 min, which remained notably lower than that for pure N$_2$ (ratio ≈ 1). This shift suggests a change in the underlying reaction mechanism, likely due to the direct formation of O radicals from H$_2$O rather than through OH radical-mediated oxidation of nitrogen species, a similar case in reaction with nitrogen-oxygen mixtures (dissociation of O$_2$).

To correlate these observations with thermal dynamics, we employed a single-bubble model to simulate peak temperatures during a 50 μs ultrasound period under different gas compositions (Fig. 4c, Supplementary Fig. 23a). Because the system operates in the high-frequency high-intensity focused ultrasound regime with sub-micrometre talc nuclei, bubble collapse remains close to spherical, validating the applicability of single-bubble kinetic modelling. Pure Ar yielded the highest average collapse temperature (>15000 K), followed by pure nitrogen (≈7300 K), while nitrogen–oxygen and nitrogen–hydrogen mixtures showed even lower values (≈5500 K). When 50% Ar was added, the temperatures increased significantly above 7500 K, explaining the observed enhancement in nitrogen fixation and the increased nitrate-to-nitrite ratio. In simulations varying nitrogen percentages in mixtures (Supplementary Fig. 23), the peak temperature remained relatively constant. However, a clear dependence of peak temperature on acoustic pressure was observed, aligning with experimental trends in pressure-dependent nitrogen fixation. These results collectively indicate that higher spike temperatures favour nitrate formation and shift product selectivity under sonochemical conditions.

Ultrasound frequency provides a second, independent control parameter. At lower frequencies, such as 530 kHz, the resonant bubble size increases, leading to more violent collapse events with higher peak temperatures but lower bubble population density. Under these conditions, we observed distinct shifts in nitrogen fixation behaviour. Comparing air and air–argon mixtures as feed gases (Supplementary Fig. 24), the total nitrogen fixation rates were 1.16 and 2.15 μM min$^{-1}$, respectively. The nitrate-to-nitrite ratio was below 0.2 for air and increased slightly above 0.2 with the addition of Ar, both significantly lower than the values obtained with the 820 kHz transducer. This indicates that lower-frequency ultrasound favours nitrite formation, consistent with a reaction environment dominated by strong but less frequent bubble collapses, where rapid quenching limits secondary oxidation steps leading to nitrate. These observations highlight that acoustic pressure and ultrasound frequency influence cavitation along two independent axes: acoustic pressure primarily affects bubble population density, altering the number of reactive microreactors; ultrasound frequency primarily affects collapse intensity, altering the temperature and radical production profile within each bubble. Together, these two parameters modulate both the rate and selectivity of nitrogen fixation through distinct and complementary mechanisms.

To quantitatively assess production, we normalised nitrogen production rates to both the effective reaction time and acoustic energy, which enables the direct comparison with other ultrasonic reactors and other nitrogen fixation methods. Based on the focused cylindrical reactor design, acoustic energy is confined within the reaction vial, improving energy utilisation. A precise 3D measurement of the acoustic pressure field was developed (Supplementary Figs. 25–27), from which the effective reaction volume was determined to be 6.16 mm$^3$ (Supplementary Fig. 26). The energy delivery into the reaction vial was then precisely calculated by a potential energy density approximation (see details and calculation examples in Supplementary Note 2).

We also calculated the energy cost per mole of fixed nitrogen and compared it with other well-developed nitrogen fixation technologies (Fig. 4d, Table 1 and Supplementary Table 3). Under optimised conditions, our energy cost is comparable to that of the classical Birkeland–Eyde process[32], ranging from 2.4 to 3.1 MJ mol$^{-1}$. While this remains higher than the highly optimised Haber–Bosch process (0.5–0.6 MJ mol$^{-1}$). In comparison, plasma-based nitrogen fixation methods[33], often exceed 5 MJ mol$^{-1}$ at ambient conditions, even with catalysts. Relative to other nitrogen-fixation technologies, particularly industrial processes, sonochemical methods currently exhibit moderate selectivity toward nitrogen products, and improvements in

energy efficiency and yield remain necessary. It is important to emphasise that the present study employs an experimental-scale reactor designed primarily to elucidate mechanistic principles rather than to optimise performance or demonstrate scalability. Further optimisation of product selectivity, reactor design, and energy efficiency based on large-scale reactor systems needs to be explored to advance cavitation-driven nitrogen fixation toward practical implementation.

Notably, in both plasma and our ultrasound-based systems, the energy cost for ammonia production is considerably higher than for NO$_x$ production. The reduction pathway to ammonium in our system requires 20.14 MJ mol$^{-1}$, consistent with the greater sensitivity of ammonia formation to the peak collapse temperature and pulse duration. These observations reinforce that direct oxidative nitrogen fixation is energetically more favourable[12], although the sluggish kinetics under mild conditions limit its application. We also note that co-products such as H$_2$, O$_2$ (gas) and H$_2$O$_2$ (aqueous) are also generated from water splitting during cavitation, similar to what is commonly observed in photochemical and electrochemical nitrogen fixation systems (Supplementary Table 3). While these pathways divert a portion of the input energy, the products themselves are energy-bearing species and separable, suggesting opportunities for integrated fertiliser–fuel co-production strategies.

## Dynamic reaction mechanisms

Despite the high temperature spikes during cavitation bubble collapse, reaches values sufficient to dissociate the nitrogen triple bond, nitrogen compounds such as ammonia and nitrate are stably formed. This raises a mechanistic question: how do these products survive, and how is chemical thermodynamics established under such transient conditions?

To elucidate these mechanisms, we performed density functional theory (DFT) calculations to assess the thermodynamics of nitrogen fixation under sonochemical conditions. As shown in Supplementary Fig. 28, under static high-temperature conditions, direct gas-phase reactions producing these nitrogen-containing species (NO, NO$_2$, and NH$_3$) are generally thermodynamically unfavourable. Therefore, the reaction energy landscape must be interpreted within a dynamically evolving thermal environment, where transient conditions can open otherwise inaccessible reaction pathways.

In classical nitrogen oxidation (e.g., combustion), the reaction proceeds via the Zeldovich mechanism (O + N$_2$ → NO + N) at temperatures around 1500 K. However, cavitation bubble collapse can reach temperatures higher than 5000 K, making direct N$_2$ dissociation feasible. As shown in Fig. 5a, at temperatures below 2000 K, indirect pathways are thermodynamically favoured, while direct dissociation is hindered by a high activation barrier (>6 eV). At 5000 K, the Gibbs free energy changes for both direct and indirect pathways converge, enabling multiple parallel reaction routes. In subsequent steps (e.g., NO to HNO$_2$ or HNO$_3$), the oxidation becomes increasingly endergonic, particularly for HNO$_3$, which requires additional oxygenation. This supports the observation that HNO$_2$ is the dominant nitrogen oxide product formed directly within collapsing bubbles. Furthermore, NO appears as a stable intermediate even at high temperatures, indicating its favourable formation. Its high reactivity with OH radicals or H$_2$O can result in fixed nitrogen compounds.

This thermal behaviour also governs NH$_3$ synthesis. As shown in Fig. 5b, the free energy for forming intermediates such as N$_2$H$_2$ increases with temperature, eventually becoming less favourable than direct N$_2$ dissociation. However, the subsequent hydrogenation steps are highly exothermic and require much lower temperatures. Because NH$_3$ decomposes above ≈773 K, continuous high temperatures would prevent its accumulation. Therefore, product formation is only viable if the system undergoes temperature quenching (dashed lines in Fig. 5b), as in cavitation collapse. This rapid quenching stabilises the

**Table 1 | Summary of nitrogen fixation rates in this work**

| Frequency (kHz) | Gas | Liquid | | Production rate ($\mu$M min$^{-1}$) | Production rate ($\mu$mol h$^{-1}$) | Energy cost (MJ mol$^{-1}$) | Energy-normalised production (mol kWh$^{-1}$) |
|---|---|---|---|---|---|---|---|
| 820 | Air | Water | Total N | 27.70 | 4.16 | 5.36 | 0.67 |
| | | | Ammonium | 0.46 | 0.069 | 323.71 | 0.011 |
| | | | Nitrite | 17.54 | 2.63 | 8.47 | 0.42 |
| | | | Nitrate | 9.51 | 1.43 | 15.63 | 0.23 |
| | 50% air, 50% Ar | Water | Total N | 39.67 | 5.95 | 3.75 | 0.96 |
| | | | Ammonium | 0.23 | 0.034 | 647.42 | 0.0056 |
| | | | Nitrite | 24.75 | 3.71 | 6.00 | 0.60 |
| | | | Nitrate | 14.59 | 2.19 | 10.18 | 0.35 |
| | Air | 0.05 M H$_2$SO$_4$ | Nitrate | 16.89 | 2.53 | 8.80 | 0.41 |
| | 80% N$_2$, 20% H$_2$ | Water | Ammonium | 7.38 | 1.11 | 20.14 | 0.18 |
| 530 | 50% air, 50% Ar | Water | Total N | 34.13 | 5.29 | 9.21 | 0.39 |
| | | | Ammonium | 0.29 | 0.044 | 1100.08 | 0.0033 |
| | | | Nitrite | 28.57 | 4.43 | 11.00 | 0.33 |
| | | | Nitrate | 5.24 | 0.81 | 60.00 | 0.060 |
| | Air | Water | Total N | 18.41 | 2.85 | 17.07 | 0.21 |
| | | | Ammonium | 0.095 | 0.015 | 3300.24 | 0.0011 |
| | | | Nitrite | 14.60 | 2.26 | 21.52 | 0.17 |
| | | | Nitrate | 3.81 | 0.59 | 82.51 | 0.044 |

The transducer operated at 820 kHz with a burst period of 2 ms and 100 cycles per burst, corresponding to a 6.1% duty cycle. The acoustic peak pressure is 6.77 MPa. The transducer operated at 530 kHz with a burst period of 3 ms and 100 cycles per burst, corresponding to a 6.3% duty cycle. The acoustic peak pressure is 6.24 MPa. All these reaction liquids have 1 mg mL$^{-1}$ talc, and the production rate is normalised to the effective reaction time

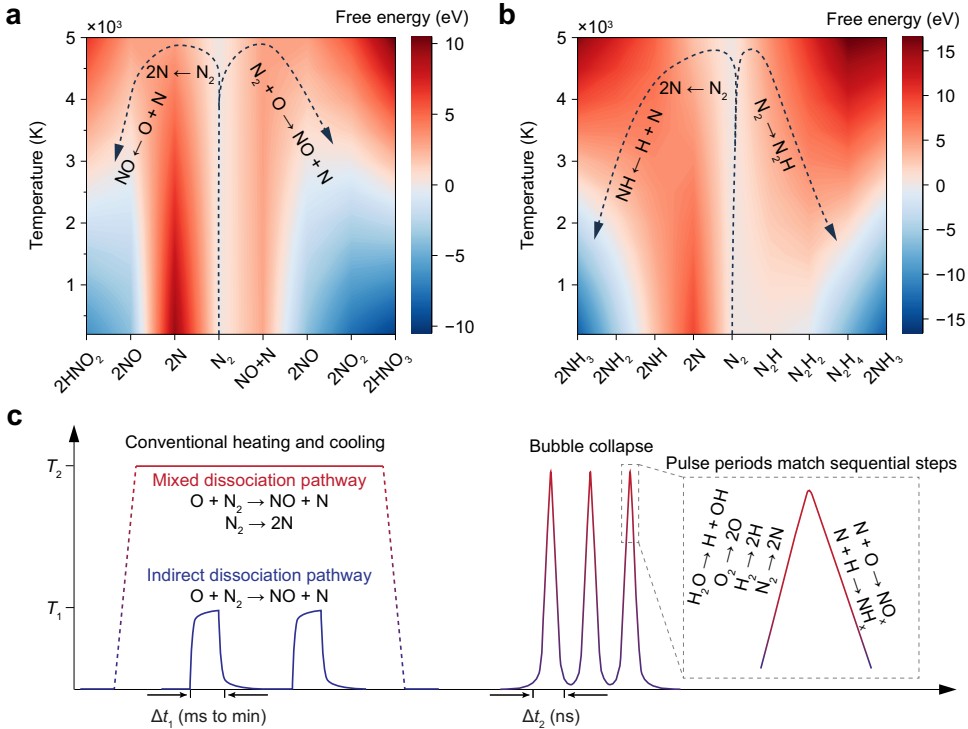

**Fig. 5 | Mechanistic investigation of sonochemical nitrogen fixation under varying temperature conditions. a** Gibbs free energy profile for nitrogen oxidation pathways: direct N$_2$ dissociation leading to HNO$_2$ formation (left) and indirect dissociation via the Zeldovich mechanism leading to HNO$_3$ formation (right). The reactants are O or OH radicals. **b** Gibbs free energy profiles for nitrogen reduction (H radicals) pathways: direct N$_2$ dissociation (left) and indirect stepwise hydrogenation routes (right). Dashed lines denote the optimal dynamic energy trajectory under temperature quenching. **c** Schematic comparison of temperature–time profiles in conventional heating/cooling systems and cavitation bubble collapse. $T_1$ represents moderate thermal conditions (1500 K), enabling indirect nitrogen activation, while $T_2$ denotes transient high temperatures (5000 K) achieved during cavitation, enabling direct N$_2$ dissociation.

products and inhibits reverse reactions, enabling the preservation of $NH_3$ and other species.

To further validate this concept, we modelled some possible oxidation and reduction pathways at both room and elevated temperatures (Supplementary Fig. 29). At low temperatures, indirect mechanisms dominate due to the high dissociation energy of $N_2$. At 5000 K, however, multiple pathways become thermodynamically accessible, supporting the notion that transient high-temperature spikes facilitate $N_2$ activation, while subsequent cooling allows product stabilisation. This dynamic separation of activation and stabilisation steps is what distinguishes sonochemical nitrogen fixation from steady-state thermal systems.

To summarise, the key advantage of cavitation chemistry lies in its ability to generate transient high temperatures (Fig. 5c). Conventional thermal systems can ramp to moderate temperatures (≈1500 K) over milliseconds to minutes, enabling only indirect dissociation pathways or catalytic reactions. Even a system capable of maintaining constant high temperatures (e.g., ≈5000 K) would not enable product formation, due to the unfavourable thermodynamics of stabilising nitrogen-containing species. Instead, cavitation enables nanosecond-scale heating and cooling, matching the timescale of sequential reaction steps and decoupling the thermodynamic and kinetic limitations. Thus, short-pulse, transient high-temperature environments promote energy utilisation and selective nitrogen product formation.

Another critical aspect is the dynamic behaviour of the cavitation bubbles, which act as microscale, transient reactors. These gas-phase domains are temporarily isolated from the liquid phase but maintain continuous mass exchange. During bubble expansion, dissolved gases are rapidly taken up; upon collapse, the newly formed reaction products are expelled into the surrounding liquid. This cyclical exchange allows for the dynamic progression of chemical reactions that are otherwise inaccessible under static conditions, further enhancing the nitrogen fixation.

## Discussion

In this study, we systematically investigated nitrogen fixation via acoustic cavitation, enabled by the transient conditions generated during ultrasonic bubble collapse. Our findings show that sonochemical nitrogen activation occurs predominantly through gas-phase reactions inside collapsing bubbles, where local temperatures can exceed 5000 K and heating/cooling rates approach $10^{12}$ K s$^{-1}$. Under these conditions, molecular nitrogen becomes sufficiently activated to undergo direct reaction with oxygen, hydrogen or water vapour, yielding nitrite, nitrate, and ammonium in the absence of any catalyst.

The tunability of product selectivity highlights the chemical versatility of cavitation-driven nitrogen fixation. Nitrogen–hydrogen mixtures favour ammonium formation, whereas nitrogen–oxygen systems selectively produce nitrates, especially under acidic conditions. Cavitation nuclei lower the cavitation threshold and improve bubble dynamics reproducibility, which provides stable reaction kinetics for long-term reactions. Ar doping modifies the collapse temperature, modulates the chemistry pathway and eventually shifts nitrate-to-nitrite ratios. Moreover, frequency and acoustic pressure act along largely independent axes, collapse intensity versus bubble population density, providing a mechanistic framework for engineering sonochemical reactors.

Isotopic labelling and mass spectrometry confirm continuous water dissociation under sonication, generating reactive radicals that participate in nitrogen fixation, though with lower yield compared to direct mechanisms, but affect the product distribution. Thermodynamics calculations reveal a dynamic temperature-dependent mechanism: while direct $N_2$ dissociation becomes thermodynamically accessible at high temperatures, subsequent steps such as hydrogenation or oxidation require rapid quenching to stabilise nitrogen products. This supports a two-step cavitation-driven

mechanism, high-temperature activation followed by low-temperature stabilisation, offering selective control beyond the limitations of conventional steady-state catalysis. Through calibrated acoustic field measurement, we quantified the acoustic energy input during sonochemical reactions. The resulting optimal energy cost for nitrogen fixation through the oxidative pathway is 3.75 MJ mol$^{-1}$.

Overall, this work further resolves where and how nitrogen chemistry occurs during bubble collapse and highlights the intrinsic suitability of cavitation for nitrogen fixation. The coexistence of oxidative and reductive pathways within a single collapse event arises from rapid quenching and efficient product transfer into the liquid phase, distinguishing this mechanism fundamentally from steady-state reaction systems. Together, these findings establish cavitation-driven non-equilibrium chemistry as a distinct mechanistic platform and underscore the broader opportunities offered by transient environments for chemical synthesis.

## Methods

### Materials

Commercial talc powder (average particle size ≈10 μm) was purchased from Sigma-Aldrich. HPLC-grade water for gradient analysis was obtained from Fisher Chemical. Gases, including argon (99.998%), nitrogen (99.998%), hydrogen (99.99%), oxygen (99.6%), and compressed air (21% ± 0.5% $O_2$ in $N_2$), were supplied by BOC. Prior to use, all gases were purified through a double-trap system consisting of a first trap containing 0.1 M sodium hypochlorite solution (5% active chlorine, Thermo Scientific) in 0.1 M sodium hydroxide solution (≥98%, Sigma-Aldrich), followed by a second trap containing 0.05 M sulphuric acid solution (95%–97%, Sigma-Aldrich). Nitrite concentrations were determined using a commercial modified Griess reagent (Sigma-Aldrich). Nitrate measurements involved the use of hydrochloric acid (≈37%, Fisher Chemical) and sulphamic acid (≥99%, Sigma-Aldrich) solutions. Ammonium concentrations were measured via the salicylate method, utilising salicylic acid (Scientific Lab Supply), sodium citrate tribasic dihydrate (≥99%, Sigma-Aldrich), sodium nitroprusside dihydrate (≥99%, Sigma-Aldrich), sodium hypochlorite (5% active chlorine, Thermo Scientific), and sodium hydroxide (≥98%, Sigma-Aldrich). For isotopic labelling experiments, deuterium oxide ($D_2O$, 99.9 at.% D), was purchased from Sigma-Aldrich, and $^{15}N_2$ gas (>98%) was purchased from Cambridge Isotope Laboratories.

### Characterisation

The powder X-ray diffraction was measured by Bruker D8 Advance Eco Cu sourced diffractometer with a fluorescence filtering LYNXEYE XE-T detector, the voltage is 40 kV, and current is 25 mA. The morphology was characterised by a Zeiss Merlin Compact Field Emission Gun (FEG)-SE, accelerating voltage is 3 kV. The EDX mapping was performed by the Oxford Instruments Ultim Xtreme X-ray detector.

### Sonochemical reaction system

In the sonochemical reactor[34], a cylindrical tube piezoelectric ceramic transducer operating at 820 kHz was used to generate ultrasound. A waveform generator (Wavestation 2052, Teledyne LeCroy) was employed to produce a pulsed sine wave, which was subsequently amplified by a radiofrequency amplifier (AR 125A250 RF Amplifier). In a typical experiment, the ultrasound parameters were set to 100 cycles per burst with a burst period of 2 ms, corresponding to a duty cycle of 6.1% (see details in Supplementary Note 1). The transducer and the reaction vial are cooling down by circulating cooling water (10 °C). This temperature is chosen to balance the gas solubility and reaction rate in the bulk of the solution.

### Cavitation probability measurement

Cavitation probability as a function of peak negative acoustic pressure was quantified at two driving frequencies, 820 kHz and 530 kHz, for

three sample media: a 1 mg mL$^{-1}$ talc suspension in air-saturated water, air-saturated water alone, and degassed water. At each pressure level, a 100-cycle tone burst was delivered repetitively to the sample ($N_{burst}$ = 1000). The burst-repetition period was 2 ms at 820 kHz and 3 ms at 530 kHz. Acoustic emissions resulting from each burst were recorded using an immersion transducer (Olympus, Japan VU-V384, centre frequency 3.5 MHz) operated as a Passive Cavitation Detector (PCD) and positioned beneath the sample. The PCD signal was pre-amplified (Stanford Research Systems SR445A) and digitised using a TiePie Handyscope HS5 oscilloscope at a sampling rate of 100 MHz.

### Acoustic potential energy method

Acoustic waveforms were measured using a calibrated Fibre-optic Hydrophone Systems sensor (FOHSv2, Precision Acoustics), with a fibre and active element diameter of 125 μm and 10 μm, respectively. The Fibre-optic Hydrophone (FOH) was mounted onto a 3D positioning system (3-Stepping Motor Controller, Velmex VXM) and used to scan in 0.2 mm increments along the x and y axes (normal to waveform propagation), and either 0.5 mm or 1 mm increments along z (concentric to radially converging field) for the 530 kHz and 820 kHz fields, respectively.

### Nitrogen fixation experimental details

In a typical nitrogen fixation experiment, 2.5 mL HPLC-grade water containing the desired concentration of talc was loaded into a polyethylene tube (11 mm diameter) and sealed with a PTFE tee connector. The talc suspension was prepared at least one day in advance to allow any potential soluble impurities to leach into the solution prior to use. Blank control samples were collected and analysed, confirming that ammonium, nitrite, and nitrate remained below the detection limit, indicating that talc does not introduce detectable nitrogen-containing contaminants. The schematic diagram is shown in Supplementary Fig. 1. A double three-way valve arrangement allows switching between a continuous flow mode, in which the feed gas passes through the reactor and into an acid trap, and a closed mode, in which the feed gas first saturates the liquid and is then isolated for sonication. The feed gases were purified using two gas scrubbers connected in series. For the continuous flowing reaction, a 1/16-inch PTFE tube is inserted into water to keep gas purging in water (flow rate: 10 mL min$^{-1}$). The minimal flow ensures stable and reproducible cavitation due to the cavitation activity is strongly localised in the focal zone. Upon completion of the sonochemical reaction, the liquid was retrieved and filtered using a syringe filter (Fisherbrand, Nylon, 25 mm, 0.2 μm pore size) to remove talc particles. The filtered liquid was then analysed using specific colourimetric reagents. For gas-phase product analysis, a quadrupole mass spectrometer (HPR-20, Hiden Analytical Ltd) with a secondary electron multiplier detector was used (ionisation energy 70 eV).

### Quantification of nitrogen-containing products and hydrogen peroxide

Standard solutions were prepared by serial dilutions. The 100 ppm-N ammonia solution was purchased from Thermal Scientific (Orion 100 ppm Ammonia as Nitrogen (N) Standard). The standard nitrite and nitrate solution was prepared from homemade 1000 ppm solutions. Potassium nitrite (ACS reagent, ≥96.0%, Sigma-Aldrich) and potassium nitrate (ACS reagent, ≥99.0%, Sigma-Aldrich) were used for the standard solution preparation. The hydrogen peroxide standard solutions (1000 ppm) were prepared from a standard 35% solution (stabilised, SAFC). Then, measuring specific volumes of solutions and using volumetric flasks to prepare ammonium, nitrate, nitrite and hydrogen peroxide solutions from 1 to 0.05 ppm. The concentrations of nitrogen-containing products were quantified using a UV-vis spectrometer (Shimadzu UV-2600) equipped with a multicell sample compartment. Three soluble nitrogen species, ammonium, nitrite, and nitrate, were detected in this study, and their corresponding calibration curves are provided in Supplementary Fig. 2. Nitrate measurements were performed using a quartz micro-cuvette, whereas ammonium and nitrite were analysed using polystyrene micro-cuvettes, each with a standard 10 mm optical path length.

Ammonium concentrations were determined using the salicylate method[35]. In a typical procedure, 100 μL 0.55 M sodium hydroxide colouring solution (5.0 wt.% salicylic acid and 5.0 wt.% sodium citrate), 20 μL 0.01 g mL$^{-1}$ sodium nitroprusside dihydrate aqueous solution, and 20 μL oxidising solution (0.75 M sodium hydroxide in sodium hypochlorite solution) were sequentially added to 2 mL sample solution. After mixing, the reaction mixtures were allowed to stand for 1 h at room temperature, and the absorbance was measured at 680 nm.

Nitrite concentrations were measured using a modified Griess method (following Sigma-Aldrich kit protocols). For each measurement, 2 mL sample solution was mixed with 1 mL Griess reagent. After standing for 15 min at room temperature, the absorbance was recorded at 540 nm.

Nitrate concentrations were determined by direct absorbance measurement after chemical pretreatment[28]. Specifically, 2 mL sample solution was mixed with 40 μL 1 M hydrochloric acid solution and 40 μL 0.8 wt.% sulphamic acid solution. The absorbance at 220 nm ($A_1$) and 275 nm ($A_2$) was recorded, and the final absorbance was calculated by $A = A_1 - 2A_2$.

Hydrogen peroxide concentrations were determined by the potassium titanium oxide oxalate (PTO) method[28]. To prepare the reagent, 1.5 g PTO was dissolved in 40 mL water to obtain a 0.1 mol L$^{-1}$ PTO solution. For the sample measurement, take 2 mL the sample, and add 100 μL reagent. After standing for 5 min, the absorbance at 385 nm was measured.

### Isotopic products measurement

The ammonium products from $^{15}N_2$ were measured by Bruker Avance III 700 with a $^1H$ inverse TCI cryoprobe equipped with a 16.44 T magnet; the frequency for $^1H$ spectra is 699.9 MHz. The solution is prepared by mixing 50 μL 0.05 M H$_2$SO$_4$, 50 μL 2% 3-(trimethylsilyl)−1-propanesulfonic acid sodium salt aqueous solution, with 900 μL solution filtered after reaction. The nitrate products were measured by the BioAccord LC-MS System. Reversed-phase chromatography was performed on an ACQUITY I-Class PLUS UPLC System (Waters, Milford, MA, USA) coupled to an ACQUITY RDa mass spectrometer (Waters, Milford, MA, USA) equipped with an ESI probe, in negative ion mode.

### DFT calculations of free energy profile

The computations for molecules were performed by means of spin-polarised density functional theory (DFT) methods using the DMol$^3$ code[36,37]. The meta-generalised gradient approximation (m-GGA) and the M06-L exchange-correlation functional[38] were employed. The All Electron core treatment was adopted, in which all electrons are included in the calculation. The double numerical plus polarisation (DNP)[36] was chosen as the basis set for calculation. The convergence criteria are: (1) a self-consistent field (SCF) tolerance of 10$^{-6}$ arb.u.; (2) an energy change of 10$^{-6}$ Ha; (3) a max force tolerance of 0.002 Ha Å$^{-1}$; (4) a maximal displacement tolerance of 0.005 Å. To ensure high-quality results, the real-space global orbital cutoff radius was chosen as high as 3.1 Å and thermal smearing with a width of 0.005 Ha was applied to the orbital occupation to speed up convergence. The Multiplicity was set according to the specific spin state of radicals.

### Single bubble collapse simulation

The temperature of single spherical collapsing bubbles in water saturated with different dissolved gas mixtures was numerically investigated with the Keller-Miksis model[39]. The Keller-Miksis equation was solved with a fourth-order Runge-Kutta method. The initial radius of the bubble was set to 5 μm in thermal, mechanical, and chemical

equilibrium with the liquid. The heat capacity ratio of the gas mixture was computed using the following: $\gamma_{N2} = 1.47$, $\gamma_{O2} = 1.4$, $\gamma_{Ar} = 1.67$, $\gamma_{H2} = 1.41$. The acoustic pressure at the focal point was obtained experimentally. Simulations required further inputs such as the water vapour pressure at 11 °C (liquid temperature) of 2065 Pa, water density of 997 kg m$^{-3}$, water viscosity of 0.89 mPa s$^{-1}$, water surface tension of 0.07275 N m$^{-1}$, and speed of sound in water of 1480 m s$^{-1}$.

## Data availability

The data that support the findings of this study are available from the corresponding authors upon request. Unprocessed raw data are provided via Figshare[40]. Source data are provided with this paper.

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

## Acknowledgements

J.K. acknowledges the Engineering and Physical Sciences Research Council (Grant Reference EP/W012316/1) and EPSRC UKRI Impact Acceleration Account Award (Grant Reference EP/X525777/1). The

authors gratefully acknowledge the late Prof. Edman Tsang's group and the Department of Chemistry at the University of Oxford for providing experimental and characterisation facilities.

## Author contributions

X.P. and J.K. conceived the concept and designed the experiments. X.P. and D.B.P. carried out the main experiments. D.B.P. performed the simulation of the bubble collapse process. Q.L. performed the DFT calculations. L.M. characterised the acoustic field of transducers, cavitation noise and developed the energy calculation methods. M.S. and P.S. helped with the reactor design, modelling and improvement, and economic analysis. Y.Q. performed SEM and EDX experiments. X.P., D.B.P. and J.K. analysed the data and wrote the manuscript. All authors discussed the data and commented on the manuscript.

## Competing interests

The authors declare no competing interests.
