## [Transparent Peer Review file · Nature Communications]

Mechanistic insights into the non-equilibrium thermodynamics of nitrogen fixation via acoustic cavitation

Corresponding Author: Professor James Kwan

Version 0:

Reviewer comments:

Reviewer #1

(Remarks to the Author)

This manuscript by Pan et al reports a novel sonochemical approach to nitrogen fixation by exploiting the transient extreme conditions generated during acoustic cavitation. The authors demonstrate that both oxidative (nitrate/nitrite) and reductive (ammonium) products can be obtained without catalysts, with selectivity tuned by feed gas composition, cavitation agents, and solution conditions. They combine experimental results, isotopic labelling, and thermodynamic modelling to propose that ultrafast thermal cycling during bubble collapse enables pathways otherwise inaccessible in conventional steady-state systems. Energy-normalized yields approach those of historical electric arc methods, offering potential for decentralized, low-carbon fertiliser production.

The concept is very interesting and could be highly impactful for sustainable nitrogen chemistry, especially given the interest in green ammonia and nitrate production. On this basis, the work seems suitable for Nature Communications. I reviewed this paper along with a couple of my research group members, and here are our collated recommendations for revisions to the paper:

1. Line 130: could more details on the construction of this "closed system" be provided?
2. Line 138: The authors claim that the introduction of talc particles lowers the cavitation threshold. It would be useful if more evidence could be supplied to support this. What effect do the talc particles have on the number of nucleating sites within the samples? Perhaps acoustic data on the bubbles would shed more light?
3. Line 148: "...the ratio of nitrate to nitrite was found to depend on the pH of the solution, implying that nitrate is likely formed through a secondary oxidation of nitrite in the liquid phase, rather than being a direct product of bubble collapse". The data in Figure 1d seem to imply that the primary outcome of switching to acid is to reduce the yield of nitrite, whereas the amount of nitrate produced remains unchanged within error of the amounts with talc or with no additive. Therefore, although the ratio of nitrite to nitrate is changing, this is more to do with less nitrite forming than giving any clues as to how nitrate forms. Therefore, we think this part of the paper needs revision.
4. In lines 195–196, the authors attribute the increase in nitrogen fixation to stronger cavitation activity during bubble collapse. This explanation is somewhat vague. It is generally known that increasing the acoustic pressure increases the bubble population, while the intensity of collapse depends majorly on the frequency (which is directly related to bubble size) — a parameter that is constant in this study. Therefore, the authors should reconsider their explanation by accounting for bubble populations. Regarding the second factor mentioned in the paper here, would tests to directly measure OH concentration support the authors' claims (although the concentration of H₂O₂ was reported, it does not directly translate to OH concentration under these conditions)?
5. In lines 219–222, the authors claim that nitrate yields continued to increase with higher pulse energy, whereas nitrite production plateaued at ~6 mJ. No reason was provided for this observation. Is this the result of acoustic saturation?
6. In lines 224–232, the introduction of talc, in addition to its effect on the nucleating sites for bubble formation, could also modify the nature of bubble propagation. In fact, bubble jetting is highly possible due to the potential non-uniform collapse of bubbles in the presence of talc. Also, relating jetting bubble collapse to the peak temperature inside the bubbles would shed more light.
7. Line 248: The H₂O/D₂O ratio would be useful to know.
8. Line 273-275: No "before" image of the talc has been provided in Supplementary Figure 16, only the images of the talc after sonication. Some EDX mapping would also be nice to see if chemical composition as well as morphology remains unchanged.

9. Lines 262-263: do the authors mean nitrite to nitrate, instead of the other way round?
10. Line 524: It is unclear whether the methods presented in this section are standard or if they are taken from other research articles. Either way, they should be appropriately referenced.
11. The information summarised in Table 2 in the Supplementary information seems central to the paper's claims. It would therefore be useful we think to put this table in the main text and discuss the results in comparison to the examples in the table in more detail in the main text.

Reviewer #2

(Remarks to the Author)

The authors would like to report nitrogen fixation via acoustic cavitation, a non-equilibrium technique that balances kinetics and thermodynamics for nitrogen fixation. In this work, rapid temperature ramping and quenching contribute to the production of ammonium, nitrate, and nitrite. While this is a systematic work, its originality is not strong enough. The concept of bubble collapse, which provides high transient temperatures and cooling rates to drive nitrogen activation and conversion, has been well developed in many literatures (e.g., J. Am. Chem. Soc. 2024, 146, 14765-14775). In addition, the lack of appropriate performance evaluation (such as production yield, product selectivity, and so on) makes it difficult to demonstrate the potential for practical application of this technology. The detailed comments are as follows:

1. The catalytic activities are mostly determined in the form of "concentration (μM)". It seems "Yield (mmol/h)" is a more appropriate form to reflect the activity, because it can be directly used for comparison with other methods. Therefore, the comparison of production yield between this method and other methods (including thermochemical, photochemical, electrochemical, mechanochemical, and laser-powered nitrogen fixation) is necessary. It is crucial to understand its potential for practical application.
2. The detailed analysis of gas composition is lacking. The authors verify the existence of H_2 in the gas product. What about other possible by-products such as O_2 , NO , and NO_2 ? It is clear that nitrogen fixation via acoustic cavitation is complex and not readily controllable, many unexpected side reactions may happen. Furthermore, the generated gaseous by-products would reduce the product selectivity. Therefore, product selectivity should be accurately determined based on all liquid and gaseous products, not just considering ammonium, nitrate, and nitrite. Likewise, does this method have an advantage over other methods (thermochemistry, photochemistry, electrochemistry, and so on) in terms of product selectivity?
3. The ammonium was detected in solution. Is gaseous NH_3 detected in the gas product?
4. For the optimization experiments, Air and N_2 flow rates may have a significant influence on the catalytic activity. They should also be detailed studied.
5. Bubble collapse could provide high transient local temperatures, but the bulk temperatures of the solution seem controlled by the cooling water system. Thus, the setting temperatures are important for the experimental detail, which is, however, missing in the manuscript. In addition, do the setting temperatures affect the catalytic activity?
6. The authors set up the closed and continuous systems. Generally, industrialization requires a continuous process rather than a batch process, due to the high yield and low energy consumption. However, it is not clear whether the continuous system is superior to the closed system in this work. A detailed table comparing these two processes is necessary.
7. In the continuous process, the stability of the catalytic activity is a vital parameter. In Fig. 3, it is hard to observe its stability. Appropriate exhibition of stability is necessary, yield (mmol/h), rather than total amount, as a function of reaction time is recommended.
8. The usage of talc indeed improved the catalytic activity, but also resulted in the contamination issue. Is there a method to separate talc and N-containing products after the reaction?
9. When conducting the mechanism study, the authors rely solely on the theoretical models. The in situ characterization (such as FTIR) is necessary to determine the formation of NO^* , NH^* intermediates.
10. In Fig. 2c, the concentration of ammonium is much higher than the total concentration of nitrate and nitrite for some points. Considering it is under an acidic condition (i.e., $[\text{H}^+]$ concentration is higher than $[\text{OH}^-]$ concentration), how can it balance the excessive positive charges in the solution? Please explain this phenomenon.

Reviewer #3

(Remarks to the Author)

This manuscript presents a systematic investigation of nitrogen fixation driven by acoustic cavitation, emphasizing the gas-phase chemistry occurring during ultrasonic bubble collapse. The authors combine experimental observations with DFT calculations to demonstrate that transient high-temperature conditions ($>5000\text{ K}$) enable N_2 activation and formation of nitrite, nitrate, and ammonium in the absence of a catalyst. The study provides a mechanistic picture of sonochemical nitrogen fixation, identifying key factors such as gas composition, acidity, and cavitation agents in controlling product selectivity.

The results are noteworthy in that they integrate mechanistic insights with tunability of product formation, supported by isotopic labeling and modelling. However, the concept of sonochemical nitrogen fixation is not new.

The methodology is sound and well executed, with appropriate use of spectroscopic and computational tools. The data and analysis appear to support the proposed conclusions, though the extent of quantitative validation (e.g., yield comparison, radical quantification) could be clarified. The mechanistic interpretation is plausible and well argued. The methods seem sufficiently detailed to allow reproduction.

Recommendations:

- 1) It is not clear why the very high energy costs of nitrogen fixation by acoustic cavitation, as conducted in this study, are

discussed mainly in relation to historical electric arc processes—this comparison is made three times in the manuscript. The appropriate benchmark, as noted on line 372, should instead be the Haber–Bosch process, which requires approximately 0.5 MJ mol^{-1} , compared to about 20 MJ mol^{-1} for the acoustic cavitation process. Along these lines, a comparison of production rates and energy costs with recent developments in ammonia synthesis by electrochemically assistance using proton-conducting ceramic cells as well as by in-liquid plasma catalysis would also be of interest to the reader.

2) The manuscript is somewhat difficult to read due to the very long captions accompanying rather small figures, as well as the large number of figures placed in the Supplementary Information. A full research article format, offering more space, would likely result in a more reader-friendly layout.

In summary, the work revisits an established but fascinating phenomenon and refines our mechanistic understanding of sonochemical nitrogen fixation. While the study is of clear scientific merit, its scope and degree of novelty make it more appropriate for a specialized journal in physical chemistry or sonochemistry rather than Nature Communications.

Reviewer #4

(Remarks to the Author)

Version 1:

Reviewer comments:

Reviewer #1

(Remarks to the Author)

The authors have fully addressed my comments. Therefore, from my perspective, I see no reason not to publish the paper in its current form in Nature Communications. The work is interesting and it will be of considerable import to the community in my view.

Reviewer #2

(Remarks to the Author)

Regarding the revised manuscript entitled “Efficient Nitrogen Fixation via Non-Equilibrium Thermodynamics Induced by Acoustic Cavitation”, the reviewer appreciates the authors’ efforts to improve the manuscript quality. However, the reviewer still considers the level of innovation to be insufficient, and technology lacks convincing prospects for practical application.

1. In the authors’ explanation of the innovation of their work on Page 9, although the authors claimed the physical origin of the transient high-temperature state is different between their work and the previously reported literatures, these distinctions cannot be regarded as significant conceptual innovation or advancement sufficient to meet the standards of Nat. Commun.

2. In Table R1, it is apparent that the authors selectively highlight the limitations of other methods regardless of their strengths, which is not an objective practice from a scientific perspective.

3. In response to comment 4, the authors claimed the intrinsic reaction was implemented in a very small reaction volume (2.5 ml), with a minimized gas flow rate. If the flow is further increased, both performance and reproducibility would dramatically decrease. This demonstrates that the technology is far from being applicable for large-scale industrial production.

4. Many of the responses to the reviewers’ previous concerns remain rather superficial. For instance, in the previous comment 6, the reviewer asked for a detailed comparison between the batch and continuous systems. However, Table R2 provides no quantitative performance comparison and presents the information solely in the form of textual statements.

Reviewer #4

(Remarks to the Author)

Response letter to reviewers' comments

Manuscript ID: # NCOMMS-25-58387A-Z

General Response to Reviewers

We sincerely thank the editor and reviewers for their careful evaluation of our manuscript entitled "*Efficient Nitrogen Fixation via Non-Equilibrium Thermodynamics Induced by Acoustic Cavitation*". We fully appreciate the time and effort invested in the review process and acknowledge the constructive feedback provided. A detailed point-by-point response follows.

Reviewers' comments:

Reviewer #1 (Remarks to the Author):

This manuscript by Pan et al reports a novel sonochemical approach to nitrogen fixation by exploiting the transient extreme conditions generated during acoustic cavitation. The authors demonstrate that both oxidative (nitrate/nitrite) and reductive (ammonium) products can be obtained without catalysts, with selectivity tuned by feed gas composition, cavitation agents, and solution conditions. They combine experimental results, isotopic labelling, and thermodynamic modelling to propose that ultrafast thermal cycling during bubble collapse enables pathways otherwise inaccessible in conventional steady-state systems. Energy-normalized yields approach those of historical electric arc methods, offering potential for decentralized, low-carbon fertiliser production.

The concept is very interesting and could be highly impactful for sustainable nitrogen chemistry, especially given the interest in green ammonia and nitrate production. On this basis, the work seems suitable for Nature Communications. I reviewed this paper along with a couple of my research group members, and here are our collated recommendations for revisions to the paper:

Response: We thank Reviewer #1 for their thoughtful and constructive comments, as well as their positive assessment. We appreciate the detailed technical questions regarding cavitation behaviour, pH influence, and the effect of talc on nucleation. In response, we provide expanded details and schematic diagrams for the closed reactor system and acoustic field characterisation; Include new acoustic measurements and optical observations to quantify bubble populations and nucleation dynamics; Clarify the role of acidity and talc on product distribution, with updated data and discussion supported by additional EDX mapping. We are confident these additions will resolve all specific queries while reinforcing the mechanistic interpretation of bubble dynamics and nitrogen fixation selectivity.

1. Line 130: could more details on the construction of this "closed system" be provided?

Response: We thank the reviewer for this helpful suggestion. We agree that the description of the closed and continuous reaction configurations should be made clearer. In the revised manuscript, we will expand the Methods section and include a labelled schematic illustrating both systems.

As shown in Fig. R1, the reaction cell consists of a PTFE fitting with one gas inlet and one outlet. The inlet is connected to a 1/16-inch PTFE capillary tube that extends into the reactor solution to introduce gas. A double three-way valve arrangement allows switching between (i) a continuous flow mode, in which the feed gas passes through the reactor and into an acid trap, and (ii) a closed mode, in which the feed gas first saturates the liquid and is then isolated for sonication. The acid trap uses the same PTFE configuration to capture any gaseous nitrogen species formed under continuous operation. This arrangement enables precise control of gas composition while preventing contamination or escape of reactive gas-phase products.

We have added a schematic diagram and corresponding description to the revised Supplementary Fig. 1 and added a brief reference to this figure in the main text (page 5 and Methods).

Figure R1. (new Supplementary Fig. 1 b,c) (a) Schematic illustration of the closed and continuous flow configurations. Red lines indicate gas lines. The system incorporates two three-way valves that switch between continuous flow (gas passing through the reactor and into the acid trap) and closed mode (gas is first introduced to saturate the liquid, after which the reactor is isolated for ultrasound irradiation). The reactor cell contains a PTFE gas inlet/outlet assembly, the inlet is connected to a 1/16 inch PTFE capillary tube submerged in the solution to ensure controlled bubble introduction and uniform gas dissolution. (b) The photograph of sonoreactor system.

2. Line 138: The authors claim that the introduction of talc particles lowers the cavitation threshold. It would be useful if more evidence could be supplied to support this. What effect do the talc particles have on the number of nucleating sites within the samples? Perhaps acoustic data on the bubbles would shed more light?

Response: We thank the reviewer for this insightful suggestion. We agree that clarifying the role of talc in controlling cavitation activity will strengthen the manuscript. Fig. R2 shows the cavitation probability as a function of the peak negative pressure reached during the rarefaction phase of the ultrasonic wave (820 and 530 kHz), both in degassed water, air saturated water and water containing 1 mg/mL talc. In pure water, cavitation occurs randomly because it depends on the presence of persistent microbubbles or impurities that act as cavitation nuclei. Previous studies have reported that, in pure water, cavitation requires more than 30 MPa of negative pressure to take place (*Phys. Rev. E*, 2015, 92, 023019). Consequently, relying on such a purely stochastic process makes sonochemical reactions difficult to control and predict.

Adding talc helps initiate cavitation by introducing a fixed number of nuclei that serve as carriers of gas impurities. This lowers the cavitation threshold and reduces the energy needed for cavitation to occur. The number of bubbles generated depends directly on the number of added nuclei, which in turn is determined by the concentration of talc particles in the solution, a parameter kept constant in our experiments. When enough talc is present, the cavitation probability approaches unity, meaning that cavitation occurs with every ultrasonic pulse. Under these conditions, cavitation is no longer random, energy losses are minimised, and the results become highly reproducible. Moreover, the size distribution of the introduced nuclei is also more controlled compared to the random impurities or microbubbles naturally present in water, further enhancing the consistency and predictability of the process.

In the revised manuscript, we included a discussion in the main text (page 6) and added new data and reference in the supplementary information.

Fig. R2. (new Supplementary Fig. 5) Cavitation probability as a function of peak negative acoustic pressure in degassed water (green), air saturated water (blue) and water containing 1mg/mL talc (red), at driving frequencies of (a) 820 kHz and (b) 530 kHz. (c) Example of power density spectra used for estimating cavitation probability, averaged over 1000 bursts at 820 kHz and 6.3 MPa peak negative pressure.

3. Line 148: “...the ratio of nitrate to nitrite was found to depend on the pH of the solution, implying that nitrate is likely formed through a secondary oxidation of nitrite in the liquid phase, rather than being a direct product of bubble collapse”. The data in Figure 1d seem to imply that the primary outcome of switching to acid is to reduce the yield of nitrite, whereas the amount of nitrate produced remains unchanged within error of the amounts with talc or with no additive. Therefore, although the ratio of nitrite to nitrate is changing, this is more to do with less nitrite forming than giving any clues as to how nitrate forms. Therefore, we think this part of the paper needs revision.

Response: We thank the reviewer for this thoughtful observation. We agree that the original wording suggested a mechanistic conclusion that was not sufficiently supported by the data and therefore requires clarification. Upon re-examination, we confirm that nitrite concentration decreases significantly under acidic conditions, whereas nitrate remains approximately constant within experimental uncertainty, resulting in an increased nitrate-to-nitrite ratio. At first glance, this may appear as if acidification solely suppresses nitrite formation. However, the total fixed nitrogen decreases in both pure water and talc-assisted systems when the solution is acidified (Fig. R3).

This behaviour is consistent with the well-established salting-out effect, in which the solubility of gases decreases in the presence of ionic species (*Ultrason. Sonochem.*, 2022, 82, 105860). In our case, the addition of H₂SO₄ reduces the dissolved concentrations of both N₂ and O₂, lowering the total extent of nitrogen fixation. The key point is therefore selectivity rather than absolute concentration. Under acidic conditions, nitrite formation is more strongly suppressed than nitrate. Although nitrate increases only slightly (within error), its fractional contribution increases, leading to a higher nitrate-to-nitrite ratio, demonstrate the selectively production of nitrate.

As discussed in the manuscript, nitrogen activation occurs primarily inside cavitation bubbles, where NO is formed as a key intermediate. Nitrite and nitrate may then form either: (i) via sequential gas-phase oxidation during collapse, or (ii) through post-collapse aqueous-phase oxidation pathways. The observed pH-dependent shift is therefore best interpreted as modulation of post-collapse liquid-phase chemistry, rather than evidence of nitrate forming exclusively through secondary oxidation of nitrite.

We have modified the relevant discussion in revised manuscripts, and added a new reference #29.

Fig. R3. The comparison of products from pure nitrogen and air in water, acidic solution (0.05 M H₂SO₄) and water with talc (1 mg/mL).

4. In lines 195–196, the authors attribute the increase in nitrogen fixation to stronger cavitation activity during bubble collapse. This explanation is somewhat vague. It is generally known that increasing the acoustic pressure increases the bubble population, while the intensity of collapse depends majorly on the frequency (which is directly related to bubble size) — a parameter that is constant in this study. Therefore, the authors should reconsider their explanation by accounting for bubble populations. Regarding the second factor mentioned in the paper here, would tests to directly measure OH concentration support the authors' claims (although the concentration of H₂O₂ was reported, it does not directly translate to OH concentration under these conditions)?

Response: We thank the reviewer for this insightful comment. We agree that the relationship between acoustic pressure, bubble population, and collapse intensity requires clearer explanation.

In our study, the acoustic pressure was varied while the frequency was held constant at 820 kHz. Under fixed frequency, the collapse intensity of individual bubbles does not significantly change, but the number of active cavitation events increases with acoustic pressure, due to the pressure-dependent growth and activation of cavitation nuclei (*ChemPhysChem*, 2010, 11, 1680-1684, *Ultrasonics*, 2006, 44, e407-e410). Therefore, the observed increase in nitrogen fixation yield is attributed primarily to an increased bubble population rather than a change in the collapse strength of each bubble.

To demonstrate the role of collapse intensity independently, we performed comparison using a lower-frequency (530 kHz) transducer, which generates larger resonant bubble size and thus a stronger collapse event. As shown Fig. R4, the lower frequency selectively enhances nitrite production efficiency, resulting in only 0.2 for nitrate-to-nitrite ratio. These results suggest that lower ultrasound frequency alters the selectivity of nitrogen products, favouring nitrite formation under more intense cavitation conditions. This indicates that lower-frequency ultrasound favours nitrite formation, consistent with a reaction environment dominated by strong but less frequent bubble collapses, where rapid quenching limits secondary oxidation steps leading to nitrate. These two trends, frequency controlling collapse intensity and pressure controlling bubble population, are mechanistically

complementary but tuneable along different axes. Acoustic pressure primarily affects bubble population density, altering the number of reactive microreactors. Ultrasound frequency primarily affects collapse intensity, altering the temperature and radical production profile within each bubble. The relevant description has been added to revised manuscript (page 16).

Regarding direct measurement of OH radicals: we agree that H₂O₂ concentration does not directly reflect OH radical concentration under these conditions. However, quantitative OH radicals probing in cavitating systems is challenging, because both the formation and consumption of OH radicals occur within the collapsing bubble, and the surviving fraction that escapes into the liquid phase is not representative of the reactive concentration that participates in nitrogen fixation. In particular, when air is used, OH radicals are rapidly consumed through reactions forming nitrite and nitrate, making the net measurable aqueous OH radicals misleading. Therefore, bulk-phase OH radicals detection cannot reliably reflect the bubble-phase radical chemistry relevant to nitrogen activation.

Fig. R4. Comparison of nitrogen fixation products with 820 and 530 kHz transducers (feed gas 50% air and 50% Ar. (a) Experiments using the 820 kHz transducer were conducted with a burst period of 2 ms and a duty cycle of 6.1%, the maximum acoustic pressure is 6.77 MPa. (b) Experiments using the 530 kHz transducer were conducted with a burst period of 3 ms and a duty cycle of 6.3%, the maximum acoustic pressure is 6.24 MPa.

5. In lines 219–222, the authors claim that nitrate yields continued to increase with higher pulse energy, whereas nitrite production plateaued at ~6 mJ. No reason was provided for this observation. Is this the result of acoustic saturation?

Response: We thank the reviewer for raising this important point. Nitrite and nitrate originate from different stages of the nitrogen oxidation pathway. In particular, nitrite is formed predominantly via the intra-bubble reaction of NO with OH, while nitrate can be formed either directly in the gas phase through further oxidation of NO or through post-collapse oxidation of nitrite in the liquid phase. At low to moderate pulse energies, nitrite formation increases because both NO and OH radicals production increase with bubble collapse intensity.

However, at higher pulse energy, two effects become dominant: (i) Enhanced oxidation inside the bubble: The concentration of reactive oxygen radicals reaches a regime where intra-bubble oxidation enhances the direct production of nitrate. This limits further growth in nitrite yield. (ii) Enhanced secondary oxidation outside the bubble: Higher pulse energy produces more OH radicals, shifts the post-collapse aqueous chemistry to favour further oxidation of nitrite to nitrate (e.g., $\text{NO}_2^- + \text{H}_2\text{O}_2 \rightarrow \text{NO}_3^- + \text{H}_2\text{O}$).

Thus, the plateau in nitrite and continued increase in nitrate are consistent with a shift from primary intra-bubble formation to secondary oxidation pathways at higher pulse energies, rather than acoustic "saturation" per se.

We have revised the manuscript to clarify this mechanistic interpretation (page 8).

6. In lines 224–232, the introduction of talc, in addition to its effect on the nucleating sites for bubble formation, could also modify the nature of bubble propagation. In fact, bubble jetting is highly possible due to the potential non-uniform collapse of bubbles in the presence of talc. Also, relating jetting bubble collapse to the peak temperature inside the bubbles would shed more light.

Response: We appreciate this point. We agree with the reviewer that bubble deformation and non-uniform collapse inevitably lead to bubble jetting towards the mentioned particles. However, as shown by Supponen et al. (*Phys. Rev. E*, 2017, 96, 033114), the collapse temperature does not show any clear trend as a function of the bubble deformation. Therefore, we do not expect a particular change in bubble temperature, neither in turn on the development of all cited chemical reactions.

However, we do believe that the energy at collapse is affected, as suggested by Supponen et al. in the article above. As reported, the energy at collapse decreases with the anisotropy parameter (indicative of the bubble deformation).

In our case, given the small size of the cavitating agents (talc particles) and the relatively high driving frequency (> 500 kHz), we expect the occurrence of small bubbles with resonance diameters in the order of 10's of microns. These eventually detach and depart from the particles, as shown by Kwan et al. in a similar work (*Phys. Rev. E*, 2015, 92, 023019). Consequently, the bubble anisotropy parameter remains small (*J. Fluid Mech.* 2016, 802, 263-293). The bubble then collapses in the weak jets regime, whereby the collapse energy is similar to the case of a spherically collapsing bubble.

Finally, we could imagine a decrease in process efficiency and chemical conversion related to energy-related matter at bubble collapse, but temperature there is expected to remain stable.

We have added description to clarify this point (page 15).

7. Line 248: The H₂O/D₂O ratio would be useful to know.

Response: We appreciate this suggestion. Specifically, we used a 3:2 volume ratio of H₂O to D₂O. In a typical experiment, 1.0 mL of 99.9 atom% D₂O was added to 1.5 mL of H₂O to form the reaction medium. This ratio ensures sufficient deuterium incorporation for isotopic tracing while maintaining similar cavitation dynamics and gas solubility to the H₂O-only system. The exact ratio and preparation steps are now reported in the caption of Fig. 3.

8. Line 273-275: No "before" image of the talc has been provided in Supplementary Figure 16, only the images of the talc after sonication. Some EDX mapping would also be nice to see if chemical composition as well as morphology remains unchanged.

Response: We thank the reviewer for this helpful clarification. The SEM image of talc before sonication was previously included in Supplementary Fig. 4, but we agree that presenting it alongside the post-sonication image would facilitate clearer comparison. We have therefore moved both "before" and "after" SEM images into a single comparative panel in the revised Supplementary Fig. 4.

In addition, following the reviewer's suggestion, we have conducted EDX elemental mapping of talc before and after sonication. As shown in Fig. R5, the elemental distributions of Mg, Si, and O remain uniform, and the Si/Mg ratios are unchanged, confirming that talc retains its chemical composition and morphology during sonication. No nitrogen signals were detected in the spectra or mapping,

ruling out the possibility of nitrogen contamination. These results confirm that talc functions purely as a physical cavitation nucleation agent rather than as a chemically active or catalytic species.

Fig. R5. (new Supplementary Fig. 18) Energy-dispersive X-ray spectroscopy (EDX) of pristine talc (a-c) and after 2 h sonication (d-f). For the pristine talc, element atomic ratios are N 0%, O 61.32%, Mg 16.05%, Si 19.89%, Cu (Substrate) 2.74%. For the talc after 2h sonication, N 0%, O 54.37%, Mg 15.35%, Si 17.83%, Cu (Substrate) 12.45%.

9. Lines 262-263: *do the authors mean nitrite to nitrate, instead of the other way round?*

Response: We thank the reviewer for noting this ambiguity. We confirm that the intended expression is the nitrite-to-nitrate ratio, not the reverse. We have corrected it in the manuscript. In the continuous-flow configuration, the continuous refreshing of dissolved gases reduces the accumulation of gas products, which stabilises the nitrite-to-nitrate ratio by limiting back-conversion or secondary oxidation pathways.

10. Line 524: *It is unclear whether the methods presented in this section are standard or if they are taken from other research articles. Either way, they should be appropriately referenced.*

Response: We appreciate this request for clarification. We clarify that the colorimetric assays used for quantifying ammonium, nitrite, nitrate, and hydrogen peroxide follow standard analytical protocols, and we have now added new references (#35 and #38) to the Methods section to explicitly acknowledge their established origins.

11. The information summarised in Table 2 in the Supplementary information seems central to the paper's claims. It would therefore be useful we think to put this table in the main text and discuss the results in comparison to the examples in the table in more detail in the main text.

Response: We thank the reviewer for this constructive suggestion. We agree that the benchmarking data summarised are central to positioning this work within the broader nitrogen fixation landscape. Due to the limitation of pages, we choose to only present our results in main text accompanying by the comparison between different industrial nitrogen fixation methods. Nevertheless, we have expanded the discussion to directly compare our energy-normalised yields and selectivity against representative nitrogen fixation systems (pages 16-18).

Reviewer #2 (Remarks to the Author):

The authors would like to report nitrogen fixation via acoustic cavitation, a non-equilibrium technique that balances kinetics and thermodynamics for nitrogen fixation. In this work, rapid temperature ramping and quenching contribute to the production of ammonium, nitrate, and nitrite. While this is a systematic work, its originality is not strong enough. The concept of bubble collapse, which provides high transient temperatures and cooling rates to drive nitrogen activation and conversion, has been well developed in many literatures (e.g., J. Am. Chem. Soc. 2024, 146, 14765-14775). In addition, the lack of appropriate performance evaluation (such as production yield, product selectivity, and so on) makes it difficult to demonstrate the potential for practical application of this technology. The detailed comments are as follows:

Response: We thank Reviewer #2 for the thoughtful comments. While we agree that transient high temperatures generated during bubble collapse are well established, our work introduces a mechanistically distinct kinetic pathway for nitrogen fixation. Previous work only emphasised the role of "extreme" temperature, our findings emphasise that the rate of temperature ramping and subsequent rapid quenching is a key determinant of nitrogen activation under non-equilibrium conditions, and that cavitation provides a highly suitable environment for exploring chemical reactions that operate on comparable timescales. Moreover, although frameworks such as the Zeldovich mechanism are widely used to rationalise nitrogen chemistry at high temperature, the critical role of water in modulating reaction pathways during bubble collapse has not previously been elucidated. Our results demonstrate that the reaction proceeds primarily through gas-phase activation inside collapsing bubbles, rather than through bulk aqueous radical oxidation or catalyst-assisted routes commonly assumed in earlier sonochemical nitrogen fixation studies. This conclusion is supported by isotopic labelling, systematic variation of gas composition /cavitation-agents/solutions, single-bubble modelling, and energy-scaling behaviour. Therefore, our study not only establishes an energy-efficient reaction system but also provides a systematic mechanistic analysis from a non-equilibrium chemistry perspective.

With regard to the study cited by the reviewer (*J. Am. Chem. Soc.* 2024, 146, 14765–14775), we agree that both works are conceptually grounded in the thermodynamic requirement of high temperature for $\text{N}\equiv\text{N}$ bond activation. However, the physical origin of the transient high-temperature state in that study is fundamentally different. Laser-induced bubble chemistry is dominated by plasma formation and optical ionisation, where chemical activation is driven by photophysical excitation rather than chemistry during inertial collapse. Furthermore, although the laser pulse width is on the nanosecond scale, the effective bubble environment persists for $\sim 100\text{--}400\ \mu\text{s}$ due to plasma and thermal relaxation, producing reaction conditions that differ significantly from the adiabatic, rapidly quenched, high-pressure gas-phase environment generated in acoustic cavitation. In contrast, our ultrasound-driven system relies solely on inertial collapse, enabling precise adjustment of collapse intensity and product selectivity through acoustic pressure, duty cycle, and gas composition, and does not involve direct plasma effects.

To our knowledge, this work provides the first experimental evidence that nitrogen activation in acoustic cavitation proceeds predominantly through gas-phase collisions inside the bubble, and that product selectivity can be tuned by controlling bubble collapse dynamics. In response to the reviewer's suggestion regarding performance evaluation, we now include expanded discussion and benchmarking of production yield and energy cost, comparing our method with other methods. These comparisons demonstrate that our method is competitive among non-thermal fixation strategies, particularly for oxidative nitrogen fixation under ambient conditions.

1. The catalytic activities are mostly determined in the form of "concentration (μM)". It seems "Yield (mmol/h)" is a more appropriate form to reflect the activity, because it can be directly used for comparison with other methods. Therefore, the comparison of production yield between this method and other methods (including thermochemical, photochemical, electrochemical, mechanochemical,

and laser-powered nitrogen fixation) is necessary. It is crucial to understand its potential for practical application.

Response: We thank the reviewer for this important point. Our initial choice to report concentrations (μM) rather than yields (mmol/h) was motivated by the observation that nitrogen fixation under acoustic cavitation does not always proceed linearly with reaction time, particularly during the first 10-20 minutes of operation, where gas dissolution, bubble nucleation dynamics, and initial radical equilibria are still stabilising. Under these conditions, directly converting concentration to production rate by simply scaling to mmol/h could lead to misleading or overestimated yield values (especially when we only run the reaction for short durations). Another point we need to emphasise is that the time we use is the total time for reaction, which ultrasound only irradiates for a 6.1% of the total time (pulse mode). When comparing with other technologies, we also need to take it into account.

For this reason, our standard methodology uses a fixed reaction duration of 30 minutes for condition comparison, and we only report the product concentration. For production rates, we only report these values in our long-duration experiments (0-2 h), where the system reaches a time-invariant steady state. We use linear fitting to get the average yield value. In these cases, the rates are reported in $\mu\text{M}/\text{min}$, which can be directly converted to mmol/h because the liquid volume (2.5 mL) is constant and explicitly stated. We also note that yield-normalised metrics commonly used in catalytic ammonia synthesis (e.g., mmol/g/h or $\text{mmol}/\text{cm}^2/\text{h}$) (*Nat. Rev. Methods Primers*, 2021, 1, 56) are not directly applicable here, as the talc added in our experiments does not act as a catalyst but rather as a cavitation agent, and therefore does not influence reaction energetics through surface-mediated pathways. So, yield normalised to catalyst loading are not applicable here, as talc functions solely as a cavitation nucleus rather than as an active catalyst.

Importantly, because the underlying energy input mechanism is electrical driving of ultrasound rather than catalytic turnover, we use energy-normalised nitrogen fixation metrics (MJ/mol , mol/kWh) for cross-technology benchmarking. These metrics allow direct performance comparison with industrial–Haber Bosch process and other emerging processes. We note that the precise quantification of acoustic energy is challenging and we use established industry standards for defining the acoustic beam intensity from measured 3D scans of the acoustic field to calculate the acoustic potential energy. All these data can be found in Table 1 in the manuscript and notes in supplementary information. The use of acoustic energy is to account for potential innate inefficiencies in our prototype reactor that may not be present in commercial systems (e.g., electrical and acoustic impedances, attenuation losses, etc.).

In response to the reviewer's request, we now additionally report the steady-state yield rates in $\mu\text{mol/h}$ in Table 1 and have expanded the discussion in the revised manuscript (page 15) to explicitly compare production rates across nitrogen fixation strategies, which will favour the direct comparison between broad fields.

2. The detailed analysis of gas composition is lacking. The authors verify the existence of H_2 in the gas product. What about other possible by-products such as O_2 , NO , and NO_2 ? It is clear that nitrogen fixation via acoustic cavitation is complex and not readily controllable, many unexpected side reactions may happen. Furthermore, the generated gaseous by-products would reduce the product selectivity. Therefore, product selectivity should be accurately determined based on all liquid and gaseous products, not just considering ammonium, nitrate, and nitrite. Likewise, does this method have an advantage over other methods (thermochemistry, photochemistry, electrochemistry, and so on) in terms of product selectivity?

Response: We thank the reviewer for raising this important point. We have now expanded and clarified our gas-phase product analysis. During bubble collapse, water undergoes pyrolysis, generating H, OH, and O radicals that can recombine to form H_2 , H_2O_2 , and O_2 . To quantify gaseous products, we analysed the gases products after reaction using quadrupole mass spectrometry (QMS). As shown in Fig. R6, a clear signal at $m/z = 32$ confirms the presence of O_2 . H_2 was observed as

previously reported. We did not detect NO or NO₂ in the gas phase, this is consistent with their known high solubility and rapid hydrolysis under aqueous conditions. Thus, nitrogen oxides are retained in the liquid phase rather than released as gases.

To ensure that additional nitrogen-containing intermediates were not overlooked, we quantified hydrazine (N₂H₄) and hydroxylamine (NH₂OH) using established colourimetric methods: the 8-quinolinol assay for hydroxylamine (*Nat. Commun.* 2024, 15, 1535) and the Watt–Chrisp method for hydrazine (*Commun. Chem.* 2021, 4, 10). In both cases, the concentrations were below the detection limit (<0.05 ppm). We therefore confirm that ammonium, nitrite, and nitrate constitute the nitrogen-containing products under our reaction conditions. Importantly, the sum of concentrations of these products increase linearly with reaction time, demonstrating both steady-state operation and accurate collection of all nitrogen-derived products.

Regarding selectivity, we agree that gas-phase H₂ and O₂ formation reflects parallel radical chemistry. However, these species arise from the same radical pool (H, OH, O) that drives nitrogen activation inside collapsing bubbles, and therefore represent mechanistically coupled rather than purely competitive pathways. In other words, water-derived radicals are not side-reactions to be suppressed, but necessary participants enabling both oxidative (NO_x formation) and reductive (NH_x) nitrogen fixation pathways.

Finally, in response to the reviewer's question about selectivity relative to other nitrogen fixation routes, we now include in the revised manuscript a comparative table (Table R1, **Supplementary Table 3**) summarising yield, selectivity, reaction pathway (oxidative or reductive), and operational challenges across thermochemical, electrochemical, photochemical, mechanochemical, plasma, and sonochemical systems. This comparison highlights that our system uniquely achieves selective nitrogen oxidation directly from air under ambient bulk temperature and without catalysts, in contrast to electrochemical and photochemical systems which typically needs reactive intermediates and show limited selectivity and low reaction rate in water-based environments.

Figure R6. Mass spectroscopy results of gas products after 30-minute reaction in closed. The water is saturated by the feed gas (N₂, H₂, and Ar), run the reaction for 30 minutes and open the valve connected to capillary sampling inlet connected to quadruple mass spectrometry.

Table R1 (new Supplementary Table 3.) Comparison between nitrogen-fixation approaches (In addition to Haber-Bosch, other methods are aqueous solutions based)

Process	Reaction Type	Typical Conditions	Typical Selectivity	Yield/Rate	Challenges	Reference
Haber–Bosch (industrial)	Reduction to NH ₃	400–500 °C, 150–300 atm, Fe catalysts, H ₂ from fossil sources	Very high selectivity (NH ₃)	Very high (tonne-scale continuous), 15% per-pass conversion	Extremely energy-intensive, centralised infrastructure, reliance on fossil H ₂ , CO ₂ emissions.	1,2
Electrocatalytic N ₂ reduction (aqueous)	Reduction to NH ₃	Ambient condition, transition metal catalysts	Very low selectivity (NH ₃ vs. H ₂ evolution competition), low faradaic efficiency (<~1–20%)	Low rate 10 ⁻⁵ –10 ⁻³ μmol/cm ² /s	Poor selectivity due to competing HER, stability of catalysts.	3,4
Photocatalytic N ₂ reduction (aqueous)	Reduction to NH ₃	Ambient condition, semiconductor photocatalysts + light	Very low selectivity	Low rate < 1–1000 μmol/g/h	Weak N ₂ activation, low quantum efficiency, poor light absorption/utilisation.	5
Plasma reduction (gas-phase or plasma-liquid hybrid)	Reduction to NH ₃	Ambient condition, non-thermal plasma activating N ₂ + hydrogen source (H ₂ or H ₂ O)	Moderate, NH ₃ often mixed with NO _x	mg-g/h NH ₃ scale depending on reactor	Competition between reduction and oxidation, NH ₃ decomposition in plasma, efficiency strongly reactor-dependent.	6,7
Mechanochemical N ₂ activation	Reduction to NH ₃	Ball milling with metal catalysts under N ₂ , solid–solid contact activation	Moderate selectivity (depends on catalyst/stoichiometry)	Gram-scale reported but from batch reaction, discontinuous	Scale-up limited, mechanical wear, energy cost per mole often high, solid handling complexity.	8
Our sonochemical reaction (N ₂ -H ₂ mixtures, aqueous)	Reduction to NH ₃	Ambient conditions, high-intensity ultrasound	High selectivity, tuneable by gas composition and catalyst surface	Up to ~0.18 mol N fixed per kWh, linear energy scaling observed	Mechanism control still developing, requires optimisation of bubble dynamics and gas–liquid transfer.	This work
Birkeland–Eyde	Oxidation to NO _x and conversion to HNO ₃	Electric arc at >3000 K, atmospheric air	Moderate selectivity (HNO ₃)	Low energy efficiency (~60 MWh per tonne HNO ₃)	Extremely electricity-intensive, temperature quenching, historically abandoned for energy reasons.	9
Electrochemical Oxidation of N ₂ (aqueous)	Oxidation to nitrite/nitrate in solution	Ambient conditions, catalyst/radical pathways	Moderate selectivity (indirect reaction pathway), low faradaic efficiency (<20%)	35–600 μmol/h/g _{cat} range, often mixed NO ₂ ⁻ /NO ₃ ⁻	Requires strong oxidising potentials, competing OER, poor selectivity control.	10,11
Plasma oxidation (gas-phase or plasma-liquid hybrid)	Oxidation to nitrate/nitrite in solution	Ambient conditions, plasma discharge in air or on water interface	Moderate selectivity (mixture of NO ₂ ⁻ , NO ₃ ⁻ , etc)	Wide range (ppm to % scale) depending on reactor and absorption	Competing radicals, product overoxidation, unstable discharge conditions.	12,13
Our sonochemical oxidation system (air / Air–Ar mixtures)	Oxidation to nitrate/nitrite in solution	Ambient conditions, high-intensity ultrasound	Moderate selectivity (mixture of NO ₂ ⁻ , NO ₃ ⁻ , etc), suitable for liquid fertiliser	Up to ~0.67 mol N fixed per kWh, energy-normalised yield significantly exceeds prior ultrasound reports	Further mechanistic quantification needed to improve energy efficiency and H ₂ O ₂ competition.	This work

References for Table R1

1. Humphreys, J., Lan, R. & Tao, S. Development and recent progress on ammonia synthesis catalysts for Haber–Bosch process. *Adv. Energ. Sust. Res.* **2**, 2000043 (2021).
2. Jennings, J. R. *Catalytic ammonia synthesis: fundamentals and practice*. (Springer Science & Business Media, 1991).
3. Ren, Y. *et al.* Strategies to suppress hydrogen evolution for highly selective electrocatalytic nitrogen reduction: challenges and perspectives. *Energ. Environ. Sci.* **14**, 1176–1193 (2021).
4. Qing, G. *et al.* Recent advances and challenges of electrocatalytic N₂ reduction to ammonia. *Chem. Rev.* **120**, 5437–5516 (2020).
5. Tang, X. C. *et al.* Recent advances in photocatalytic nitrogen fixation based on two - dimensional materials. *ChemCatChem* **16**, e202401355 (2024).
6. Sakakura, T., Murakami, N., Takatsuji, Y. & Haruyama, T. Nitrogen fixation in a plasma/liquid interfacial reaction and its switching between reduction and oxidation. *J. Phys. Chem. C* **124**, 9401–9408 (2020).
7. Zhao, X. & Tian, Y. Sustainable nitrogen fixation by plasma-liquid interactions. *Cell Rep. Phys. Sci.* **4**, 101618 (2023).
8. Han, G.-F. *et al.* Mechanochemistry for ammonia synthesis under mild conditions. *Nat. Nanotechnol.* **16**, 325–330 (2021).
9. Rouwenhorst, K. H., Jardali, F., Bogaerts, A. & Lefferts, L. From the Birkeland–Eyde process towards energy-efficient plasma-based NO_x synthesis: a techno-economic analysis. *Energ. Environ. Sci.* **14**, 2520–2534 (2021).
10. Chen, S. *et al.* Direct electroconversion of air to nitric acid under mild conditions. *Nat. Syn.* **3**, 76–84 (2024).
11. Wan, H., Bagger, A. & Rossmeisl, J. Limitations of electrochemical nitrogen oxidation toward nitrate. *J. Phys. Chem. L* **13**, 8928–8934 (2022).
12. Bradu, C., Kutasi, K., Magureanu, M., Puač, N. & Živković, S. Reactive nitrogen species in plasma-activated water: generation, chemistry and application in agriculture. *J. Phys. D Appl. Phys.* **53**, 223001 (2020).
13. Bruggeman, P. J. *et al.* Plasma–liquid interactions: a review and roadmap. *Plasma Sources Sci. T.* **25**, 053002 (2016).

3. The ammonium was detected in solution. Is gaseous NH₃ detected in the gas product?

Response: We thank the reviewer for this helpful question. To determine whether ammonia is released in the gas phase during sonication, we incorporated an acid trap downstream of the reaction vessel in the continuous-flow configuration (Fig. R7a). The trap contained 0.05 M H₂SO₄, such that any volatilised NH₃(g) would be quantitatively captured and converted to NH₄⁺ (aq). After reaction, the contents of both the reaction vial and the acid trap were analysed using the indophenol blue method.

As shown in Fig. R7b, the ammonium concentration in the acid trap remained below the detection limit (<0.05 ppm) even after 2 hours of continuous operation, whereas the reaction vial exhibited clear accumulation of ammonium. This indicates that no measurable NH₃ is released into the gas phase during the reaction. Instead, ammonia formed inside cavitation bubbles dissolves rapidly upon bubble collapse due to its high Henry's law constant and strong acid–base buffering in the aqueous phase. The absence of NH₃ in the gas phase is consistent with prior reports of sonochemically generated NH₃ in water.

This confirms that all detectable ammonia remains in the liquid phase, and the quantified ammonium represents the total NH₃ produced. We added related description in revised manuscript (page 12).

Figure R7. (part of new Supplementary Fig.17) (a) Schematic of the continuous-flow sonochemical system with downstream acid trap for capturing gaseous NH_3 . (b) UV-vis absorbance curves from ammonium assays of the reaction vial (solid line) and acid trap (dashed line), showing detectable ammonium only in the reaction solution. Conditions: feed gas is 80% N_2 and 20% H_2 , frequency is 820 kHz, and duty cycle is 6.1%.

4. For the optimization experiments, Air and N_2 flow rates may have a significant influence on the catalytic activity. They should also be detailed studied.

Response: We thank the reviewer for this important point. We agree that gas flow rate is a key parameter in large-scale acoustic reactors where gas-liquid mass transfer and bubble seeding can significantly influence cavitation dynamics. In the present work, our aim was to establish intrinsic reaction behaviour under controlled and reproducible conditions within a small reaction volume (2.5 mL). For this reason, the gas flow rate was intentionally minimised to maintain a steady saturated state while avoiding turbulence and bubble coalescence effects that would disrupt the focused acoustic field.

Our reactor employs a high-intensity focused ultrasonic beam (Fig. R8a–b), and the cavitation activity is strongly localised in the focal zone. When gas flow rates were increased beyond the minimal saturation rate, we observed that large bubbles introduced by sparging scattered and defocused the acoustic energy field, resulting in reduced nitrogen fixation efficiency and decreased reproducibility. Therefore, we saturated the solution prior to sonication and maintained only a low, constant flow rate to preserve gas composition without disturbing the acoustic field.

We agree, however, that gas flow becomes a major optimisation parameter for larger-scale or non-focused reactors, where the bubble population distribution interacts differently with the acoustic field. Indeed, previous work (*Ultrason. Sonochem.* 2022, 90, 106214) has demonstrated that controlled gas sparging can increase cavitation activity in large reactors by modifying the spatial extent of the active zone. We would like to clarify that: (i) In the present focused microreactor, minimal flow ensures stable and reproducible cavitation. (ii) In larger reactors, gas flow optimisation is expected to be beneficial and will be a key focus of future scale-up work.

We have added a schematic of the reactor geometry and focal region (Fig. R8) and clarified the gas saturation procedure and flow conditions in the Methods section (page 23).

Fig. R8. (a) Schematic of the focused ultrasound reactor showing transducer geometry and gas inlet configuration. (b) 3D acoustic field mapping of the focal region, illustrating localisation of cavitation activity.

5. Bubble collapse could provide high transient local temperatures, but the bulk temperatures of the solution seem controlled by the cooling water system. Thus, the setting temperatures are important for the experimental detail, which is, however, missing in the manuscript. In addition, do the setting temperatures affect the catalytic activity?

Response: We thank the reviewer for raising this point. As noted in the Methods section, the reactor is water-jacketed at 10 °C, and direct thermocouple measurement inside the reaction vial shows a stable bulk liquid temperature of ~11 °C during sonication. We will include the exact measured temperature in the revised manuscript.

The effect of temperature can be considered from two perspectives: (i) Gas solubility: Lower bulk temperature increases the solubility of both N₂ and O₂, increasing their concentration in the bubbles prior to collapse. Since nitrogen activation occurs inside the bubbles, higher dissolved gas concentration favours nitrogen fixation. Therefore, low bulk temperature is beneficial. (ii) Bulk-phase reaction kinetics: Bulk temperature influences only post-collapse reactions involving longer-lived species (e.g., OH recombination, nitrite oxidation). These reactions occur after bubble collapse and do not determine the primary N₂ activation pathway.

Thus, the dominant chemistry of nitrogen fixation takes place within ultrahigh temperature microreactors (the bubbles), and is not governed by bulk heating. We have clarified this explicitly in the Methods section (page 23).

6. The authors set up the closed and continuous systems. Generally, industrialization requires a continuous process rather than a batch process, due to the high yield and low energy consumption. However, it is not clear whether the continuous system is superior to the closed system in this work. A detailed table comparing these two processes is necessary.

Response: We thank the reviewer for the comments. I agree that industrialisation requires a continuous process, so we only reported reaction rate data with flow system, which is related to comment 1. Herein, we didn't directly compare the closed and flow systems, as there is no

potential to apply a batch mode for practical applications. In sonochemical reactions, gas bubbles as nuclei are necessary to get efficient cavitation, which can be created by continuously sparging gases. In our work, we reported some reactions in batch mode, is to use it as a model to investigate the reaction mechanism, which help us identify the relationship between nitrite and nitrate. We do not think closed system could be an option for practical applications as it will be degassed after certain time of sonication. The comparison is shown in Table R2. The continuous flow system is the configuration relevant for scalable operation, and therefore all reported reaction rates and energy-efficiency benchmarks are based on the continuous system.

Table R2 Comparison table summarising the differences between closed system and continuous flow system.

Feature	Closed System (Batch)	Continuous Flow System
Purpose	Mechanistic investigation	Practical operation & rate quantification
Gas Supply	Static, pre-saturated	Constant gas refresh, controlled composition
Cavitation Stability	Limited, degas easily	Stable due to continuous bubble seeding
Suitability for Scale-Up	Not suitable	Suitable and preferred

We clarified this distinction in the revised manuscript (page 11).

7. In the continuous process, the stability of the catalytic activity is a vital parameter. In Fig. 3, it is hard to observe its stability. Appropriate exhibition of stability is necessary, yield (mmol/h), rather than total amount, as a function of reaction time is recommended.

Response: We agree that demonstrating stability is important. While our system does not involve a "catalyst", stability here refers to whether the cavitation intensity and bubble population remain constant over time. To address this explicitly, we performed new cavitation - probability measurements (new Supplementary Fig. 5), which confirm that talc effectively enhances cavitation activity and renders it stable and controllable throughout operation. We also conducted additional material characterisation (new Supplementary Fig. 18), demonstrating that talc maintains both structural and chemical integrity under sonication, confirming that it functions solely as a cavitation agent rather than as a reactive or degradable species.

Unlike heterogeneous catalytic systems, where continuous product sampling can be performed during a single run, our approach involves independent experiments conducted for different reaction durations. The product concentrations obtained from each experiment were then plotted as a function of reaction time.

This stability of the process is evidenced by the linear time-dependent accumulation of total fixed nitrogen, as shown in Figs. 3 and 4. Across all 2-hour continuous-flow experiments (Air and N₂-H₂ systems), the summed nitrogen product concentrations increase linearly with reaction time, with coefficients of determination $R^2 > 0.998$. This confirms that the reaction rate remains constant throughout the experiment and that the measured nitrogen products are complete and accurately quantified.

In the revised manuscript, we added R^2 results for all the linear fitting (Caption of Figs. 3, 4).

8. The usage of talc indeed improved the catalytic activity, but also resulted in the contamination issue. Is there a method to separate talc and N-containing products after the reaction?

Response: We confirm that no nitrogen-containing species are introduced by the talc. A blank talc suspension was prepared under identical sonication-free conditions and analysed. Ammonium, nitrite, and nitrate were all below detection limit. We have provided this clarification in revised manuscript (page 23).

Additionally, talc is insoluble and is removed after reaction by a 0.2 μm PTFE syringe filter, as stated in the Methods. No talc components were detected in the filtrate.

9. When conducting the mechanism study, the authors rely solely on the theoretical models. The in situ characterization (such as FTIR) is necessary to determine the formation of NO^ , NH^* intermediates.*

Response: We appreciate the reviewer's interest in intermediate identification. We note that the reaction pathway in this system proceeds via gas-phase radical chemistry inside collapsing cavitation bubbles, rather than through surface-bound intermediates (NO^* , NH^*) that are characteristic of heterogeneous catalytic systems. The relevant species (N, O, H, NO, NH, OH) are formed under transient high-temperature conditions ($>5000\text{ K}$) and have lifetimes on the sub-nanosecond scale, making direct in situ spectroscopic detection in the bulk liquid infeasible. Nevertheless, the formation of NO radicals under ultrasonic cavitation has been experimentally demonstrated in previous work using spin-trapping methods such as the sodium N-methyl-D-glucamine dithiocarbamate iron(II) complex ($(\text{MGD})_2\text{Fe}^{2+}$), (*J. Phys. Chem.* 1996, 100, 17986-17994).

We performed ex-situ ATR-FTIR on post-reaction solutions and observed bands corresponding to NO_2^- and NO_3^- , which supports the proposed reaction network but does not allow direct detection of radicals.

Fig. R9. The FT-IR spectra of products after 2h reaction with $\text{N}_2\text{-H}_2$ and $\text{N}_2\text{-O}_2$ mixtures, the concentration of N_2 is 80%.

10. In Fig. 2c, the concentration of ammonium is much higher than the total concentration of nitrate and nitrite for some points. Considering it is under an acidic condition (i.e., $[\text{H}^+]$ concentration is higher than $[\text{OH}^-]$ concentration), how can it balance the excessive positive charges in the solution? Please explain this phenomenon.

Response: We thank the reviewer for noting this. The primary nitrogen species formed during bubble collapse are neutral molecules (NH_3 , HNO_2 , HNO_3). These species are protonated or

deprotonated only after dissolution in water, e.g., $\text{NH}_3 + \text{H}^+ \rightarrow \text{NH}_4^+$, $\text{HNO}_2 \rightarrow \text{NO}_2^- + \text{H}^+$, $\text{HNO}_3 \rightarrow \text{NO}_3^- + \text{H}^+$. Thus, electroneutrality is maintained because the acid–base reactions in the solution provide the corresponding counterions, no net charge imbalance occurs.

Reviewer #3 (Remarks to the Author):

This manuscript presents a systematic investigation of nitrogen fixation driven by acoustic cavitation, emphasizing the gas-phase chemistry occurring during ultrasonic bubble collapse. The authors combine experimental observations with DFT calculations to demonstrate that transient high-temperature conditions (>5000 K) enable N₂ activation and formation of nitrite, nitrate, and ammonium in the absence of a catalyst. The study provides a mechanistic picture of sonochemical nitrogen fixation, identifying key factors such as gas composition, acidity, and cavitation agents in controlling product selectivity.

The results are noteworthy in that they integrate mechanistic insights with tunability of product formation, supported by isotopic labeling and modelling. However, the concept of sonochemical nitrogen fixation is not new.

The methodology is sound and well executed, with appropriate use of spectroscopic and computational tools. The data and analysis appear to support the proposed conclusions, though the extent of quantitative validation (e.g., yield comparison, radical quantification) could be clarified. The mechanistic interpretation is plausible and well argued. The methods seem sufficiently detailed to allow reproduction.

Response: We thank Reviewer #3 for their positive assessment of the mechanistic framework, experimental execution, and isotopic and modelling support. We also appreciate the recommendations regarding benchmarking and manuscript presentation. We agree that sonochemical nitrogen fixation has been studied before, however, we respectfully clarify that the mechanistic basis underlying the process has remained unresolved. Prior studies have generally interpreted nitrogen fixation under ultrasound through analogy to high-temperature combustion (e.g., the Zeldovich mechanism), without accounting for the ultrafast thermal cycling and non-equilibrium kinetics unique to cavitation.

Our study systematically investigated the reaction steps in the cavitation system and emphasise the role of the short pulse, which really distinguishes sonochemical nitrogen fixation from any other methods, which also represents a new non equilibrium chemistry mechanism. Our work establishes and validates a gas-phase non-equilibrium activation–quenching route inside individual cavitation bubbles, which provide new understanding of nitrogen fixation through both reduction and oxidation pathway. To our knowledge, this direct radical chemistry pathway has not been described previously and addresses major mechanistic uncertainty in the field.

Recommendations:

1) It is not clear why the very high energy costs of nitrogen fixation by acoustic cavitation, as conducted in this study, are discussed mainly in relation to historical electric arc processes—this comparison is made three times in the manuscript. The appropriate benchmark, as noted on line 372, should instead be the Haber–Bosch process, which requires approximately 0.5 MJ mol⁻¹, compared to about 20 MJ mol⁻¹ for the acoustic cavitation process. Along these lines, a comparison of production rates and energy costs with recent developments in ammonia synthesis by electrochemically assistance using proton-conducting ceramic cells as well as by in-liquid plasma catalysis would also be of interest to the reader.

Response: We thank the reviewer for raising this important benchmarking point. Our system operates mainly via the oxidation pathway (N₂ → NO_x⁻) with air as feed gas, whereas Haber–Bosch proceeds via reductive hydrogenation (N₂ → NH₃). Because these two processes differ fundamentally in both thermodynamics and product form, the mechanistically appropriate industrial analogue is the Birkeland–Eyde (BE) electric arc process, which also forms NO_x

directly from air under high-temperature nonequilibrium conditions. This is why BE was used as our primary benchmark.

We acknowledge that the Haber–Bosch process is currently the most optimised nitrogen fixation technology, operating close to its thermodynamic and practical limits. In contrast, the sonochemical nitrogen fixation approach reported here is still at an early stage of development and far from being fully optimised. The system involves a large multidimensional parameter space, including ultrasonic frequency, acoustic pressure, focal zone geometry, gas composition, reaction temperature, gas flow rate and reactant delivery, among others. These parameters require systematic investigation to approach higher efficiency. We also recognise that side reactions such as water splitting and hydrogen peroxide formation divert part of the energy input away from nitrogen fixation. However, we note that such competing reactions are commonly observed in other aqueous nitrogen reduction systems, including electrochemical and photocatalytic processes. To provide a clearer contextual comparison, we have now included a new comparison table in the Supplementary Information (Supplementary Table 3), outlining key differences among major nitrogen fixation approaches, including the referred electrochemical methods and in-liquid plasma catalysis.

Overall, while our current energy efficiency does not yet match the level of the industrial Haber–Bosch process, the sonochemical pathway remains promising, with substantial room for optimisation. Importantly, the ability to drive the process using clean electricity and to produce liquid nitrogen fertilisers directly from air and water presents a compelling opportunity for decentralised, small-scale, and distributed fertiliser production. We believe that highlighting this broader context, including both oxidative and reductive nitrogen fixation pathways, will be beneficial to readers given the relevance to sustainable fertiliser manufacturing.

2) The manuscript is somewhat difficult to read due to the very long captions accompanying rather small figures, as well as the large number of figures placed in the Supplementary Information. A full research article format, offering more space, would likely result in a more reader-friendly layout.

Response: We thank the reviewer for this practical and constructive suggestion. In the revised manuscript, we have enlarged the main figures to improve clarity. We acknowledge that the manuscript contains substantial information, as our aim is to present a systematic study. To maintain a clear narrative in the main text, we have retained the essential results in the main figures and moved supporting but potentially distracting details to the Supplementary Information. We have also carefully revised the manuscript to streamline the logic and improve readability, including shortening and reorganising some sections.

In summary, the work revisits an established but fascinating phenomenon and refines our mechanistic understanding of sonochemical nitrogen fixation. While the study is of clear scientific merit, its scope and degree of novelty make it more appropriate for a specialized journal in physical chemistry or sonochemistry rather than Nature Communications.

Response: We thank the reviewer for their constructive evaluation and for recognising the scientific merit of our study. We respectfully clarify that the contribution of this work goes beyond revisiting a known sonochemical phenomenon. We clarify that our work demonstrates a direct, non-equilibrium gas-phase activation mechanism inside collapsing bubbles. This mechanistic distinction is supported by isotopic tracing, gas-composition-dependent product tuning, single-bubble cavitation modelling, DFT calculation and spatially resolved acoustic energy quantification.

To summarise the contribution of the study: (i) Resolves a longstanding mechanistic controversy in sonochemistry. (ii) Establishes a generalisable non-equilibrium activation principle with relevance beyond nitrogen. (iii) Enables direct and meaningful cross-technology energy benchmarking by developing a model to quantify acoustic energy.

Because the study combines elements of reaction engineering, sonochemistry, nitrogen activation chemistry, mechanistic modelling, and sustainable fertiliser production, we believe the work will be of broad interest to readers across catalysis, energy materials, non-equilibrium chemistry, and agricultural sustainability. For these reasons, we respectfully maintain that the manuscript aligns well with the interdisciplinary scope of *Nature Communications*.

Reviewer #4 (Remarks to the Author):

Response: We appreciate Reviewer #4's participation and their acknowledgement of the joint review process.

Response letter to reviewers' comments

Manuscript ID: # NCOMMS-25-58387B

Reviewers' comments:

Reviewer #1 (Remarks to the Author):

The authors have fully addressed my comments. Therefore, from my perspective, I see no reason not to publish the paper in its current form in Nature Communications. The work is interesting and it will be of considerable import to the community in my view.

Response: We sincerely thank Reviewer #1 for their positive evaluation of the revised manuscript. We are grateful for their careful review and for confirming that all comments have been fully addressed.

Reviewer #2 (Remarks to the Author):

Regarding the revised manuscript entitled “Efficient Nitrogen Fixation via Non-Equilibrium Thermodynamics Induced by Acoustic Cavitation”, the reviewer appreciates the authors’ efforts to improve the manuscript quality. However, the reviewer still considers the level of innovation to be insufficient, and technology lacks convincing prospects for practical application.

Response: We thank the Reviewer #2 for their continued engagement with the revised manuscript and for recognising the improvements made. We have reframed the manuscript and emphasised the mechanistic insights into the mechanism for sonochemical nitrogen fixation, which represents the distinct advancement of this work. The detailed response to all the comments is listed below.

1. In the authors’ explanation of the innovation of their work on Page 9, although the authors claimed the physical origin of the transient high-temperature state is different between their work and the previously reported literatures, these distinctions cannot be regarded as significant conceptual innovation or advancement sufficient to meet the standards of Nat. Commun.

Response: We appreciate the reviewer’s high standard for conceptual novelty. We respectfully clarify that the innovation of this work does not lie in the mere observation of transient high temperatures, but in resolving how and where nitrogen activation occurs under cavitation conditions, and why acoustic cavitation is uniquely suited to nitrogen chemistry from a thermodynamic and kinetic perspective.

Previous literature has largely interpreted sonochemical nitrogen fixation through indirect aqueous radical pathways or by analogy to combustion chemistry, without in depth experimental and theoretical investigation. Our study provides the convergent experimental and modelling evidence that nitrogen fixation proceeds predominantly through non-equilibrium activation-quenching chemistry inside collapsing bubbles. The coexistence of oxidative and reductive pathways within a single collapse event arises from rapid quenching and efficient product transfer into the liquid phase, distinguishing this mechanism fundamentally from steady-state reaction systems. This mechanistic resolution, supported by isotopic labelling, gas-composition-dependent selectivity, single-bubble modelling, and energy scaling, is not a marginal physical distinction but a foundational clarification that resolves a long-standing ambiguity in sonochemistry. We therefore believe this mechanistic resolution constitutes a foundational conceptual advance.

To more clearly present the advancement of this work, we have revised both the Introduction and Conclusions (pages 3 and 13).

2. In Table R1, it is apparent that the authors selectively highlight the limitations of other methods regardless of their strengths, which is not an objective practice from a scientific perspective.

Response: We appreciate the reviewer’s concern regarding objectivity. Our intention in Table R1 was not to selectively emphasise the limitations of other nitrogen fixation approaches, but rather to provide a transparent comparison of product selectivity, and yield metrics, which are evaluated differently across methods and are therefore inherently difficult to compare directly.

To address this concern, we have substantially revised Table R1. The updated table now focuses primarily on quantitative metrics, including reported or literature-derived yields and selectivity, rather

than qualitative assessments. Where exact values are unavailable, we explicitly state when estimates are used and provide appropriate references. In addition, the “challenges” column has been removed and the challenges only for the presented ultrasound technology is discussed. We believe this revised presentation offers a more balanced and objective comparison while still enabling readers to clearly assess how different nitrogen fixation strategies differ in mechanism, selectivity, and practical constraints.

We now include the revised table (Table R1, Supplementary Table 3) in the Supplementary information and revised manuscript accordingly (page11).

Table R1 (new Supplementary Table 3.) Comparison between nitrogen-fixation approaches (In addition to Haber-Bosch, other methods are aqueous solutions based)

Process	Reaction	Typical conditions	Selectivity	Yields	Reference
Haber–Bosch (industrial)	Reduction to NH ₃	400–500 °C, 150–300 atm, Fe catalysts, H ₂ from fossil sources	≈100% selectivity (NH ₃), 15% per-pass conversion	Tonne-scale continuous production	1,2
Electrocatalytic N ₂ reduction (aqueous)	Reduction to NH ₃	Ambient condition, transition metal catalysts	1–20% selectivity (estimated from faradic efficiency)	~10 ² –10 ⁵ μmol h ⁻¹ g _{cat} ⁻¹	3-5
Photocatalytic N ₂ reduction (aqueous)	Reduction to NH ₃	Ambient condition, semiconductor photocatalysts + light	1–10% (dominated by H ₂ production)	~1–10 ³ μmol h ⁻¹ g _{cat} ⁻¹	6
Plasma reduction (gas-phase or plasma-liquid hybrid)	Reduction to NH ₃	Ambient condition, non-thermal plasma activating N ₂ + hydrogen source (H ₂ or H ₂ O)	<20% (mixed with NO _x , H ₂ , O ₃ and H ₂ O ₂)	~10 ² –10 ³ μmol h ⁻¹ g _{cat} ⁻¹	7,8
Mechanochemical N ₂ reduction	Reduction to NH ₃	Ball milling with catalysts under N ₂ and H ₂ /H ₂ O	Up to 99% (rarely reported)	≈250 μmol h ⁻¹ g _{cat} ⁻¹	9,10
Sonochemical reaction (N ₂ -H ₂ mixtures, aqueous)	Reduction to NH ₃	Ambient conditions, high-intensity ultrasound	<80% (estimated from detectable products)	1.11 μmol h ⁻¹ , 0.18 mol kWh ⁻¹	This work
Birkeland–Eyde	Oxidation to NO _x and conversion to HNO ₃	Electric arc at >3000 K, rapid quenching, atmospheric air	No specific data (newly reported data 84% NO ₂)	~60 MWh per tonne HNO ₃	11,12
Electrochemical Oxidation of N ₂ (aqueous)	Oxidation to nitrite/nitrate in solution	Ambient conditions, catalyst/radical pathways	<30% selectivity (estimated from faradic efficiency)	~10–10 ³ μmol h ⁻¹ g _{cat} ⁻¹	13,14
Plasma oxidation (gas-phase or plasma-liquid hybrid)	Oxidation to nitrate/nitrite in solution	Ambient conditions, plasma discharge in air or on water interface	20–99% (mixture of NO ₂ ⁻ , NO ₃ ⁻ , H ₂ O ₂)	~10 ⁻¹ –10 ⁵ mg _{NOx} h ⁻¹	15-17
Sonochemical oxidation system (air/air–Ar mixtures)	Oxidation to nitrate/nitrite in solution	Ambient conditions, high-intensity ultrasound	40-70% (mixture of NO ₂ ⁻ , NO ₃ ⁻ , H ₂ O ₂)	27.7 μmol h ⁻¹ , ~0.67 mol kWh ⁻¹	This work

References for Table R1

- Humphreys, J., Lan, R. & Tao, S. Development and recent progress on ammonia synthesis catalysts for Haber–Bosch process. *Adv. Energ. Sust. Res.* **2**, 2000043 (2021).
- Jennings, J. R. *Catalytic ammonia synthesis: fundamentals and practice*. (Springer Science & Business Media, 1991).
- Ren, Y. *et al.* Strategies to suppress hydrogen evolution for highly selective electrocatalytic nitrogen reduction: challenges and perspectives. *Energ. Environ. Sci.* **14**, 1176–1193 (2021).
- Qing, G. *et al.* Recent advances and challenges of electrocatalytic N₂ reduction to ammonia. *Chem. Rev.* **120**, 5437–5516 (2020).

5. Yang, B., Ding, W., Zhang, H. & Zhang, S. Recent progress in electrochemical synthesis of ammonia from nitrogen: strategies to improve the catalytic activity and selectivity. *Energ. Environ. Sci.* **14**, 672–687 (2021).
6. Tang, X. C. *et al.* Recent advances in photocatalytic nitrogen fixation based on two - dimensional materials. *ChemCatChem* **16**, e202401355 (2024).
7. Sakakura, T., Murakami, N., Takatsuji, Y. & Haruyama, T. Nitrogen fixation in a plasma/liquid interfacial reaction and its switching between reduction and oxidation. *J. Phys. Chem. C* **124**, 9401–9408 (2020).
8. Zhao, X. & Tian, Y. Sustainable nitrogen fixation by plasma-liquid interactions. *Cell Rep. Phys. Sci.* **4**, 101618 (2023).
9. Han, G.-F. *et al.* Mechanochemistry for ammonia synthesis under mild conditions. *Nat. Nanotechnol.* **16**, 325–330 (2021).
10. He, C. *et al.* Mechanochemical synthesis of ammonia employing H₂O as the proton source under room temperature and atmospheric pressure. *ACS Sustain. Chem. Eng.* **10**, 746–755 (2022).
11. Rouwenhorst, K. H., Jardali, F., Bogaerts, A. & Lefferts, L. From the Birkeland–Eyde process towards energy-efficient plasma-based NO_x synthesis: a techno-economic analysis. *Energ. Environ. Sci.* **14**, 2520–2534 (2021).
12. Angineni, J., Reddy, P. M. K., Anga, S. & Somaiah, P. V. Nitrogen Fixation by Simple Gling Arc Plasma Reactor at Elevated Pressure for Synthesis of Aqueous Nitrogen Fertilizer. *Plasma Process. Polym.* **22**, 2400209 (2025).
13. Chen, S. *et al.* Direct electroconversion of air to nitric acid under mild conditions. *Nat. Syn.* **3**, 76–84 (2024).
14. Wan, H., Bagger, A. & Rossmeis, J. Limitations of electrochemical nitrogen oxidation toward nitrate. *J. Phys. Chem. L* **13**, 8928–8934 (2022).
15. Bradu, C., Kutasi, K., Magureanu, M., Puač, N. & Živković, S. Reactive nitrogen species in plasma-activated water: generation, chemistry and application in agriculture. *J. Phys. D Appl. Phys.* **53**, 223001 (2020).
16. Bruggeman, P. J. *et al.* Plasma–liquid interactions: a review and roadmap. *Plasma Sources Sci. T.* **25**, 053002 (2016).
17. Gromov, M. *et al.* Electrification of fertilizer production via plasma-based nitrogen fixation: a tutorial on fundamentals. *RSC Sustain.* **3**, 757–780 (2025).

3. *In response to comment 4, the authors claimed the intrinsic reaction was implemented in a very small reaction volume (2.5 ml), with a minimized gas flow rate. If the flow is further increased, both performance and reproducibility would dramatically decrease. This demonstrates that the technology is far from being applicable for large-scale industrial production.*

Response: We appreciate the reviewer’s concern about the scalability. The present reactor geometry (2.5 mL focal volume) is not designed to represent an industrially optimised system. However, this does not undermine the relevance of the chemistry demonstrated. The purpose of the current configuration is to isolate and characterise the intrinsic non-equilibrium reaction mechanism under well-defined cavitation conditions. The observation that excessive gas flow disrupts cavitation stability in a small-volume reactor is a known hydrodynamic limitation, not an intrinsic barrier to scale-up. In larger reactors, increased flow rates would be accompanied by proportionally larger cavitation-active volumes and appropriately designed acoustic fields, as is established in sonochemical reactor engineering.

Nevertheless, to ensure that the conclusions of the manuscript remain fully supported by experimental evidence, we have reframed the discussion to focus on mechanistic insight rather than scale-up claims.

The revised manuscript therefore emphasises the fundamental non-equilibrium chemistry governing nitrogen activation inside collapsing bubbles, which is independent of reactor size and represents the primary contribution of this work.

The revisions can be found in page 11, where we clarify this work is based on small scale experiments and future research needs to be explored for practical applications. All other statements that could imply large-scale applicability have been removed to avoid over-interpretation.

4. Many of the responses to the reviewers' previous concerns remain rather superficial. For instance, in the previous comment 6, the reviewer asked for a detailed comparison between the batch and continuous systems. However, Table R2 provides no quantitative performance comparison and presents the information solely in the form of textual statements.

Response: We appreciate the reviewer's continued attention to this point. In our previous response, we clarified that several of the earlier comments extended beyond the scope of the present study due to the fundamentally different reaction system. Specifically, we clarified that the batch system is used exclusively as a mechanistic probe, whereas all performance metrics (rates, energy costs, stability) are derived from continuous-flow operation. A direct "batch vs continuous" performance comparison is not provided because batch operation is not proposed as a model to derive a performance.

To directly address the reviewer's request, we have now added a quantitative comparison between batch and continuous systems using identical acoustic conditions (820 kHz transducer, identical acoustic pressure, gas composition, and liquid volume). The comparison shows that, the continuous-flow system consistently delivers higher nitrogen fixation rates than the batch system for both air and N₂-H₂ feed gases. The time linearity of product accumulation is significantly improved under continuous flow, indicating more stable cavitation activity and sustained gas availability. Batch experiments exhibit earlier deviation from linearity due to progressive gas depletion and accumulation effects, which limits their suitability for performance evaluation.

Table R2 Comparison table summarising the differences between closed system and continuous flow system.

Feature	Closed System (Batch)	Continuous Flow System
Purpose	Mechanistic investigation	Practical operation & rate quantification
Gas Supply	Static, pre-saturated	Constant gas refresh, controlled composition
Cavitation Stability	Limited, degas easily	Stable due to continuous bubble seeding
Suitability for Scale-Up	Not suitable	Suitable and preferred
Performance (nitrogen fixation rate)	0.41 $\mu\text{M}/\text{min}$ for N ₂ -H ₂ , 1.38 $\mu\text{M}/\text{min}$ for air	0.45 $\mu\text{M}/\text{min}$ for N ₂ -H ₂ , 1.69 $\mu\text{M}/\text{min}$ for air

We have included quantitative comparison in the revised manuscript (pages 7 and 8).

Reviewer #4 (Remarks to the Author):

Response: We appreciate Reviewer #4's participation and their positive comment.